# Naturally-occurring spinosyn A and its derivatives function as argininosuccinate synthase activator and tumor inhibitor

Zizheng Zou[1,2,3,4,9], Xiyuan Hu[1,9], Tiao Luo[5], Zhengnan Ming[1], Xiaodan Chen[1], Li Xia[6], Wensong Luo[1], Jijia Li[3], Na Xu[2], Ling Chen[2], Dongsheng Cao[2], Min Wen[1], Fanrong Kong[1], Kunjian Peng[1], Yuanzhu Xie[1], Xuan Li[1], Dayou Ma[2], Chuanyu Yang[7], Ceshi Chen[7], Wenjun Yi[8], Ousheng Liu[5], Suyou Liu [2✉], Junli Luo [3✉] & Zhiyong Luo [1✉]

Argininosuccinate synthase (ASS1) is a ubiquitous enzyme in mammals that catalyzes the formation of argininosuccinate from citrulline and aspartate. ASS1 genetic deficiency in patients leads to an autosomal recessive urea cycle disorder citrullinemia, while its somatic silence or down-regulation is very common in various human cancers. Here, we show that ASS1 functions as a tumor suppressor in breast cancer, and the pesticide spinosyn A (SPA) and its derivative LM-2I suppress breast tumor cell proliferation and growth by binding to and activating ASS1. The C13-C14 double bond in SPA and LM-2I while the Cys97 (C97) site in ASS1 are critical for the interaction between ASS1 and SPA or LM-2I. SPA and LM-2I treatment results in significant enhancement of ASS1 enzymatic activity in breast cancer cells, particularly in those cancer cells with low ASS1 expression, leading to reduced pyrimidine synthesis and consequently the inhibition of cancer cell proliferation. Thus, our results establish spinosyn A and its derivative LM-2I as potent ASS1 enzymatic activator and tumor inhibitor, which provides a therapeutic avenue for tumors with low ASS1 expression and for those non-tumor diseases caused by down-regulation of ASS1.

[1] Department of Biochemistry and Molecular Biology, Hunan Province Key Laboratory of Basic and Applied Hematology, Hunan Key Laboratory of Animal Models for Human Diseases, School of Life Sciences, Xiangya School of Medicine, Central South University, Changsha, China. [2] Xiangya School of Pharmaceutical Sciences, Central South University, Changsha, China. [3] The Hunan Provincial Key Laboratory of Precision Diagnosis and Treatment for Gastrointestinal Tumor, Xiangya Hospital, Central South University, Changsha, China. [4] Department of Biochemistry and Molecular Biology, Yiyang Medical College, Yiyang, China. [5] Hunan Key Laboratory of Oral Health Research & Xiangya Stomatological Hospital & Xiangya School of Stomatology, Central South University, Changsha, China. [6] Core Facility of Basic Medical Sciences, Shanghai Jiao Tong University School of Medicine, Shanghai, China. [7] Key Laboratory of Animal Models and Human Disease Mechanisms of Chinese Academy of Sciences & Yunnan Province, Kunming Institute of Zoology, Chinese Academy of Sciences, Kunming, China. [8] Department of General Surgery, The Second Xiangya Hospital, Central South University, Changsha, China. [9] These authors contributed equally: Zizheng Zou, Xiyuan Hu. ✉email: liusuyou@hotmail.com; junlluo@163.com; luozhiyong@csu.edu.cn

Argininosuccinate synthase or synthetase (ASS1) was first found in the liver and later recognized as a ubiquitous enzyme in mammals. It catalyzes the formation of argininosuccinate from citrulline and aspartate, the immediate precursor of the semi-essential amino acid arginine. Deficiency of ASS1 in patients leads to citrullinemia, an autosomal recessive urea cycle disorder characterized with marked plasma citrulline increase and punctuated hyperammonemia[1,2]. The somatic silence or down-regulation of ASS1 expression are very common in various of tumors, including melanoma, prostate cancer, breast cancer, bladder cancer, mesothelioma, pancreatic cancer, nasopharyngeal carcinoma, osteosarcomas, and myxofibrosarcomas[3-6]. Low ASS1 expression in tumor tissues is associated with poor prognosis of breast cancer and hepatocellular carcinoma patients[7,8]. The silence of ASS1 in tumors not only causes the loss of tumor suppressor role of ASS1 but also makes tumor cells be reliant on or addicted to extracellular arginine[6,9,10]. This feature has been exploited for the treatment of individuals with ASS1-deficient tumors by arginine starvation. As ASS1 is a key enzyme for arginine biosynthesis, arginine depletion results in specific cell death of ASS1-deficient cancer cells[7,11,12]. However, almost all therapeutic strategies for the treatment of tumors with low ASS1 expression ignore the tumor suppressor role of ASS1[6-8,12,13], and there are no tools currently available to reestablish the expression and/or activity of ASS1 in tumor cells.

Spinosad, a natural substance isolated from *Saccharopolyspora spinosa*, is the active ingredient in many registered pesticide products and in some drugs regulated by the US Food and Drug Administration (FDA). It is a mixture of two macrocyclic lactone compounds called spinosyn A (SPA) and spinosyn D[14]. It is both a contact and stomach poison for many caterpillar species. Owing to its favorable environmental properties such as rapid degradation, low acute mammalian toxicity[15], no carcinogenicity[16], and no cross-resistance[14], spinosad has been widely used to control a wide variety of pests, such as thrips, leafminers, mosquitoes, ants, fruit flies, as well as animal and human head lice[17-19]. It has been reported that the highly selective toxicity to insects may be related to its indirect interaction with acetylcholine receptor and gamma-aminobutyric acid (GABA) receptor of insect central nervous system[20,21]. This may not apply to mammals as it has little or low toxicity to mammals, suggesting that spinosad may function differently in mammalian cells[14-16].

Here, we demonstrate that SPA and its derivative LM-2I have potent antitumor activity, which is mediated by activation of ASS1. SPA and its derivative LM-2I specifically target and bind to ASS1 at the 97th cysteine site in tumor cells, leading to significantly enhanced ASS1 enzymatic activity and tumor inhibitory effect.

## Results

**SPA and its derivative LM-2I inhibit breast cancer cell proliferation and tumor growth**. In a cancer drug candidate screening of natural products, we found that SPA (**1**) significantly inhibited cancer cell proliferation (Fig. 1a). To confirm its role in cancer cell growth, immortalized normal human breast epithelial cells (MCF-10A and 184B5) and breast cancer (BC) cell lines, including seven triple negative breast cancer (TNBC) cell lines (MDA-MB-231, MDA-MB-468, Hs578T, BT20, BT549, HCC1806, and HCC1937), four luminal subtypes cell lines (MCF-7, HCC1500, T47D, and BT474), and two HER2 overexpression cell lines (BT474 and SKBR3), were treated with different concentrations of SPA for 48 h and the cell viability was analyzed by crystal violet staining assay. We found that SPA had mild effect on immortalized normal human breast epithelial cells while significantly inhibited the proliferation of most BC cell lines

tested in a dose-dependent manner (Fig. 1b, Supplementary Fig. S4a). Interestingly, SPA showed significantly different toxicity to different BC cells (Fig. 1b, Supplementary Fig. S4a). Similarly, SPA significantly inhibited BC cell colony formation while had much less effect on normal human breast epithelial cells (Fig. 1c).

We then synthesized a series of derivative compounds of SPA and tested their inhibitory efficacy on cancer cells by crystal violet assay and colony formation assays[22]. We found that the SPA derivative LM-2I (**2**) (Supplementary Fig. S1) had a much stronger inhibitory effect on cancer cells than SPA, while it kept low toxicity to normal human breast epithelial cells (Fig. 1d–f, Supplementary Fig. S4b).

To investigate the role of SPA and LM-2I in tumor growth inhibition, MDA-MB-231 cells were inoculated subcutaneously into BALB/c nu/nu female mice. When the average tumor size reached about 100 mm³, mice were treated with either vehicle, SPA (10 mg/kg/day), or LM-2I (5 mg/kg/day). Tumor size (Fig. 1g) and mouse body weight (Fig. 1i) were measured every two days. When the average tumor size in control group reached about 1000 mm³ (28 days), mice were sacrificed and tumors were dissected and weighted (Fig. 1h). We found that both SPA and LM-2I significantly suppressed MDA-MB-231 xenograft tumor growth in nu/nu mice (Fig. 1g, h). Notably, LM-2I showed stronger antitumor activity than SPA (Fig. 1g, h). The HE staining of tumor tissue sections showed that less tumor cell density in tumors from mice treated with SPA and LM-2I (Supplementary Fig. S5a), the expression of Ki-67 was significantly decreased in the SPA and LM-2I treated groups as compared with controls (Supplementary Fig. S5c). Importantly, the mice did not display significant changes in body weight (Fig. 1i) and histopathology of liver, heart, spleen, lung, and kidney (Supplementary Fig. S5a) in groups administrated with SPA and LM-2I. Altogether, these results indicate that SPA and LM-2I have strong antitumor activity both in vitro and in animal models, while there is no or few clear side-toxicity.

**SPA and LM-2I interact with ASS1**. SPA is one of the two components of spinosad, a natural substance that is very toxic to insects, but not to mammals[15,17-19]. The highly selective toxicity to insects is related to its indirect interaction with GABA receptors of insect central nervous system[20,21]. The low toxicity to mammals suggests that spinosad may function differently in mammalian cells[14-16]. To explore the direct targets of SPA and LM-2I in cancer cells, we prepared chemical probes for affinity purification. Studies of the relationship between structure of semi-synthesis SPA derivatives and activity indicated that the substitution at the nitrogen atom of 17-amino sugar group in SPA retained the biological activity[22], suggesting that the biotin tag could be attached to the nitrogen atom of 17-amino sugar group of SPA. In addition, the much lower active analog of SPA (SPAH, **3**) (Fig. 2a and Supplementary Fig. S2) indicated that the α, β-unsaturated ketene moiety was essential for biological activity of SPA (Fig. 2b and Supplementary Fig. S6a, b). Based on this information, we synthesized positive probe (Biotin-SPA, **4**) and negative probe (Biotin-SPAH, **5**) with biotin and alkyl linkers (Fig. 2c and Supplementary Fig. S3), respectively. Biological evaluation showed that the biotinylated positive probe had similar anti-cancer cell proliferation activity to SPA, while the biotinylated negative probe lost this activity (Fig. 2d).

We then used Biotin-SPA as a positive probe to find possible SPA-interacting proteins in cancer cells, while Biotin-SPAH used as a negative control. MCF-7 cell lysates were incubated with Biotin-SPA or Biotin-SPAH in the presence or absence of LM-2I, followed by pulling-down with streptavidin magnetic beads. The proteins bound to the magnetic beads were examined by

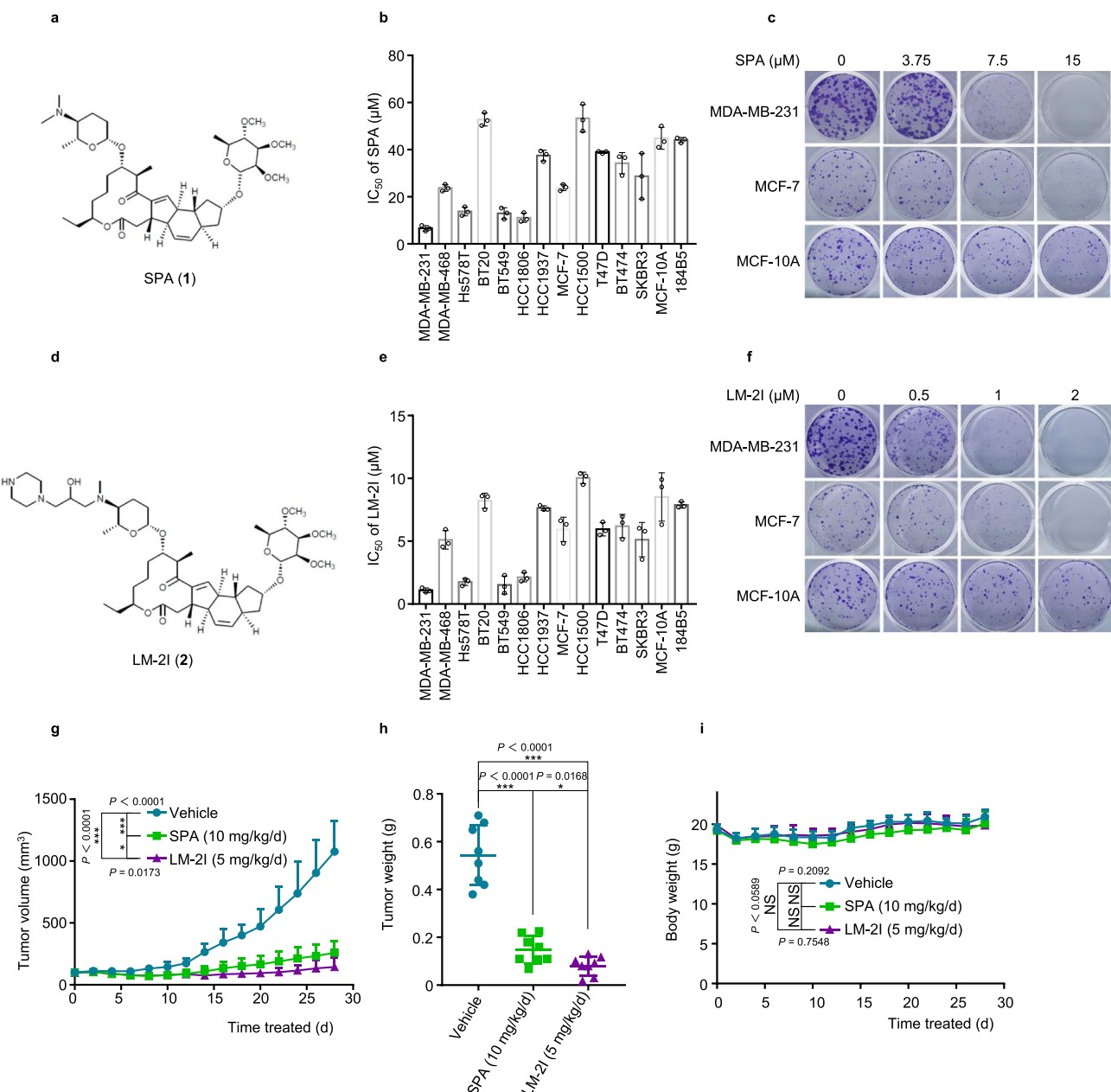

**Fig. 1 SPA and its derivative LM-2I inhibit breast cancer proliferation and growth. a** Chemical structures of SPA (**1**). **b** Crystal violet assay for the effects of SPA on breast cancer cell proliferation. IC$_{50}$ values were calculated with SPSS 18.0 (mean ± s.d., $n = 3$ biologically independent experiments). **c** Representative microscope images of colony formation for different breast cancer cell lines treated with SPA. Scale bars are 40 μM. **d** Chemical structures of LM-2I (**2**). **e** Crystal violet assay for the effects of LM-2I on breast cancer cell proliferation. IC$_{50}$ values were calculated with SPSS 18.0 (mean ± s.d., $n = 3$ biologically independent experiments). **f** Representative microscope images of colony formation for different breast cancer cell lines treated with LM-2I. Scale bars are 40 μM. MDA-MB-231 cells were inoculated subcutaneously into the right flank of nu/nu female mice. When the average tumor size reached about 100 mm$^3$, mice were treated with ether vehicle, SPA (10 mg/kg/day) or LM-2I (5 mg/kg/day). Tumor sizes (**g**) and mouse body weight (**i**) were measured every 2 days. 28 days later, mice were sacrificed and tumors were dissected and weighted (**h**). Data represent mean ± s.d. ($n = 8$ mice per group). *$P < 0.05$, ***$P < 0.001$, NS: not significant ($P > 0.05$) from two-tailed unpaired Student's $t$ tests. Source data are provided as a Source Data file.

SDS-PAGE and silver staining. We found that Biotin-SPA could specifically pull down the protein(s) with a molecular weight around 50 kDa (Fig. 2e). The binding of Biotin-SPA with the protein(s) could be competed away by unlabeled LM-2I in a dose-dependent manner (Fig. 2e), suggesting that LM-2I can also bind to the Biotin-SPA binding protein(s).

The peptide mass fingerprint analysis of this 50 kDa protein band revealed that the protein precipitated by Biotin-SPA was ASS1 (Supplementary Fig. S6c). Western blot analysis by using anti-ASS1 specific antibody confirmed that the 50 kDa protein

band precipitated by Biotin-SPA in MCF-7 cell lysates was ASS1 (Fig. 2f). Fluorescent staining of MCF-7 (Fig. 2g), MDA-MB-231 and MCF-10A (Supplementary Fig. S6d) cells preincubated with Biotin-SPA showed Biotin-SPA and ASS1 colocalized in the cytoplasm of the cells. Furthermore, the incubation time course experiments showed that the binding between ASS1 and Biotin-SPA started very quickly after Biotin-SPA was added into the recombinant ASS1 solution in vitro and increased gradually in a time-dependent manner (Fig. 2h). The addition of LM-2I interrupted the binding of ASS1 with Biotin-SPA (Fig. 2i).

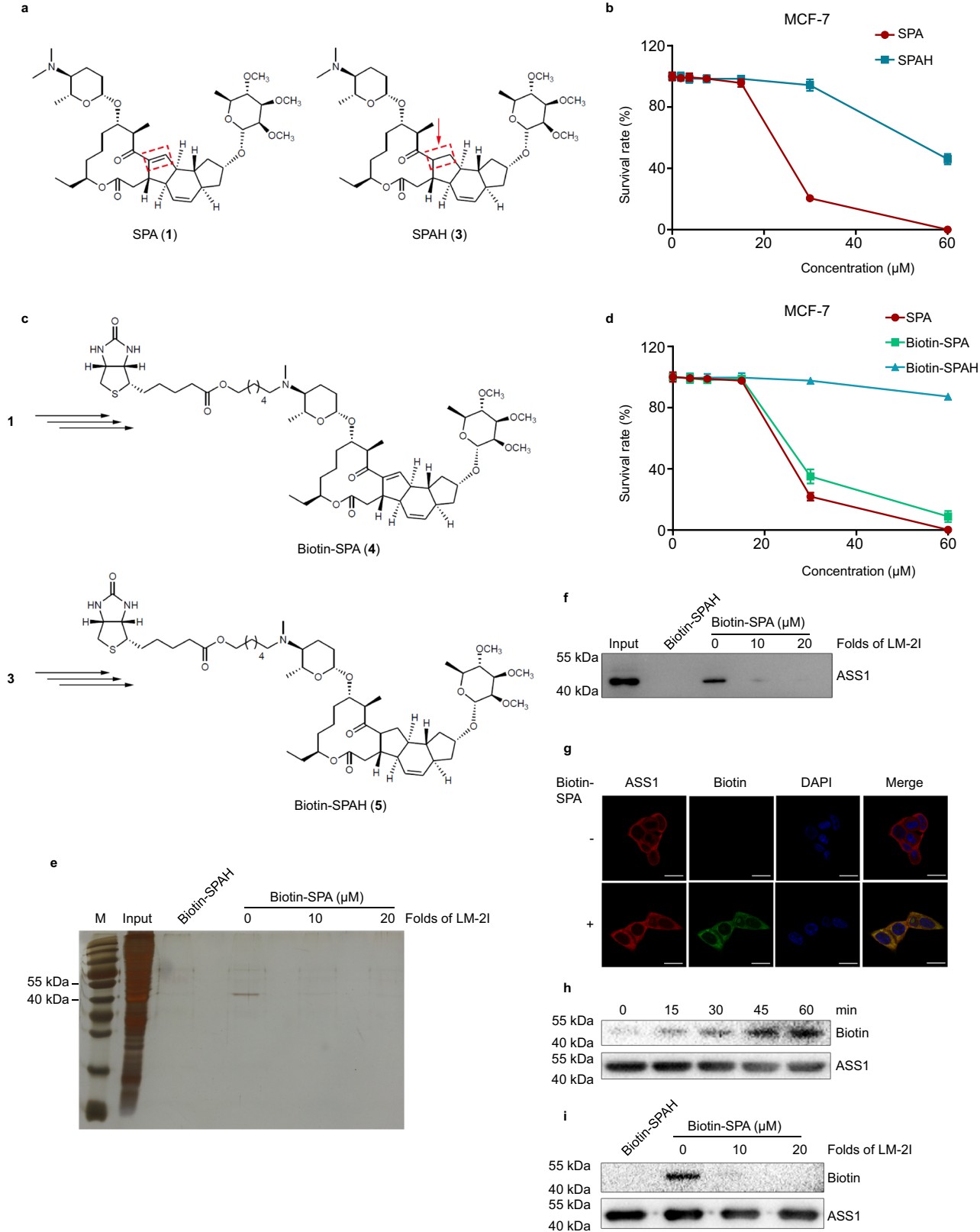

Together, these results suggest that SPA and LM-2I interact with and bind to ASS1 in cancer cells.

**Cysteine 97 (C97) of ASS1 is critical for its interaction with SPA and LM-2I.** The structure–activity relationship information suggests that the α, β-unsaturated ketene moiety in SPA is a michael acceptor which potentially captures well-positioned nucleophiles, which most likely with the thiol group of the cysteine residue of the target protein forms a covalent bond[23,24]. We found that once the C13-C14 double bond of SPA was

**Fig. 2 SPA and LM-2I directly target ASS1. a** Chemical structures of SPA (**1**) and its reductive form SPAH (**3**). **b** Crystal violet assay for the viability of MCF-7 cells treated with different concentrations of SPA and SPAH (mean ± s.d., $n = 3$ biologically independent experiments). **c** Chemical structures of biotin-labeled SPA [Biotin-SPA (**4**)] and biotin-labeled SPAH [Biotin-SPAH (**5**)]. **d** Crystal violet assay for the viability of MCF-7 cells treated with different concentrations of biotin-labeled SPA [Biotin-SPA (**4**)] and biotin-labeled SPAH [Biotin-SPAH (**5**)] (mean ± s.d., $n = 3$ biologically independent experiments). MCF-7 cell lysates were incubated with biotin or Biotin-SPA in the presence or absence of LM-2I at 4 °C overnight, followed by pulling-down with streptavidin magnetic beads. The proteins bound to the magnetic beads were separated by SDS-PAGE, followed by silver staining (**e**) or by western blot using ASS1 antibody (**f**) ($n = 3$ biologically independent experiments). **g** Fluorescent staining of MCF-7 cells incubated with or without Biotin-SPA for 4 h ($n = 3$ biologically independent experiments). Scale bars are 20 μm. **h** The recombinant ASS1 protein was incubated with Biotin-SPA for different time periods as indicated, followed by western blot using streptavidin-HRP or ASS1 antibody ($n = 3$ biologically independent experiments). **i** The recombinant ASS1 protein was incubated with Biotin-SPA in the absence or presence of LM-2I for 1 h at room temperature, followed by western blot using streptavidin-HRP or ASS1 antibody ($n = 3$ biologically independent experiments). Source data are provided as a Source Data file.

changed to a single bond (SPAH), the binding affinity between SPAH and ASS1 decreased significantly as the binding of Biotin-SPA to ASS1 could be competed away by unlabeled SPA but not SPAH (Fig. 3a), indicating that the C13–C14 double bond in SPA and LM-2I is essential for their binding to ASS1.

These results (Figs. 2b, 3a, and Supplementary Fig. S6a, b) also suggest that cysteine residues in ASS1 protein are potential important sites for the interaction between ASS1 and SPA or LM-2I. To investigate this hypothesis, we generated five mutant ASS1 recombinant proteins, in each of which only one individual cysteine of the five cysteine residues (Cys19, Cys97, Cys132, Cys331, and Cys337) was replaced by alanine or aspartic acid. Recombinant WT and mutant ASS1 proteins were incubated with Biotin-SPA for 30 min, followed by western blot using streptavidin-HRP or ASS1 antibody. We found that the mutant ASS1 protein (ASS1-C97D), in which the amino acid cysteine at 97 site was replaced by aspartic acid, lost its binding ability to Biotin-SPA, while the other ASS1 mutants maintained similar Biotin-SPA-binding affinity to WT ASS1 (Fig. 3b), suggesting that Cys97 (C97) in ASS1 is essential for the interaction between ASS1 and Biotin-SPA.

Furthermore, the recombinant ASS1 protein was incubated with SPA, LM-2I, or vehicle, and then digested with trypsin and lysC, the peptides containing cysteine residue(s) were evaluated by LC-MS/MS analysis (Fig. 3c–e). We found that the Cys97 (C97)-containing peptide shifted the molecular weight (MW) of SPA (Fig. 3c, d) or LM-2I (Fig. 3c, e). The MW of SPA mass increased by 731.46 Da while LM-2I mass increased by 859.56 Da, which matched the sum of the MW of the C97-containing peptide and SPA (Fig. 3c, d) or LM-2I (Fig. 3c, e). These results further support that SPA and LM-2I covalently modify Cys 97 but not the other four cysteines of ASS1.

To further understand the binding model of small molecule and protein, we chose LM-2I as a representative and performed a computational study using the unique crystal structure of ASS1 (Protein Data Bank (PDB) code: 2NZ2)[25], where LM-2I was manually docked to the C97 site of ASS1. In the modeled LM-2I-ASS1 complex, the Michael acceptor at C13 of LM-2I forms a C-S bond with Cys97 and forms a stable hydrogen bond with Asp182 and Phe68 (Fig. 3f).

**SPA and LM-2I promote ASS1 catalytic activity.** ASS1 catalyzes the formation of arginosuccinate from aspartate, citrulline and ATP, and is the rate-limiting enzyme for the biosynthesis of arginine in most mammal tissues. To examine whether the binding of SPA or LM-2I to ASS1 affects the enzymatic activity of ASS1, an in vitro ASS enzymatic activity assay was performed, in which recombinant ASS1 protein was incubated with different concentration of SPA (Fig. 4a) or LM-2I (Fig. 4b) at 4 °C overnight, followed by incubation with the substrates citrulline, aspartate and ATP. The ASS1 activity was assessed at different time points based on the accumulation of the product

pyrophosphate. Interestingly, we found that both SPA and LM-2I promote ASS1 enzyme activity in a dose-dependent manner, and that there was a gradual saturation trend with increasing drug concentration (Fig. 4a, left panel and 4b, left panel). We then selected the linear time period of the enzymatic reaction (1–8 min) and fitted its slope (the reaction speed) to calculate the increase ratio of the stimulus response rate of the drug at different concentrations (Fig. 4a, middle panel and 4b, middle panel). Furthermore, the Hill1 fitting of the Origin 2018 software was used, and the maximal activation ratio and the half maximal activation concentration ($AC_{50}$) of the enzymatic reaction rate were calculated (Fig. 4a, right panel and 4b, right panel). Consistent with their tumor inhibitory potency, LM-2I had stronger effect on ASS1 activation than SPA. The maximum activation rate and $AC_{50}$ of SPA and LM-2I on ASS1 were 179.68%, 18.68 μM and 398.74%, 2.09 μM, respectively. These results suggest that SPA and LM-2I interact with ASS1 and promote ASS1 enzymatic activity.

**The binding kinetics of SPA and LM-2I to ASS1.** To evaluate the binding kinetics of SPA or LM-2I to ASS1, we chose Biotin-SPA as a probe to examine the binding of SPA or LM-2I to ASS1 at various time points and at different concentrations. The time course of ASS1 binding to Biotin-SPA at various concentrations showed time-dependent saturation (Fig. 4c, top), which is consistent with an irreversible binding mechanism. The data were fit to assess the observed rate constant ($k_{obs}$) for ASS1 binding to Biotin-SPA at each concentration (Fig. 4c, bottom). Plotting these $k_{obs}$ values as a function of Biotin-SPA concentration revealed a saturation curve (Fig. 4d). These results supported a two-step reaction mechanism for the activation of ASS1 by SPA or LM-2I. The initial binding step ($K_i$) is the noncovalent reversible binding of ASS1 to SPA or LM-2I, which places the moderately reactive electrophile of SPA or LM-2I close to a specific nucleophile on the ASS1 protein. The second chemical step is irreversible, leading to specific covalent linkage ($k_{inact}$) and the complex activation. Based on this model, the $k_{inact}$ and $K_i$ for ASS1 binding with Biotin-SPA were 0.043 min$^{-1}$ and 9.011 μM, respectively. Overall, the binding process between Biotin-SPA and ASS1 is effective ($k_{inact}/K_i = 79.53$ M$^{-1}$ s$^{-1}$). In addition, we also applied Surface Plasmon Resonance (SPR) system to detect the initial noncovalent reversible binding of ASS1 to SPA (Fig. 4e) and LM-2I (Fig. 4f). Our results showed that the $K_D$ value for the binding of SPA or LM-2I to ASS1 was 0.3578 μM and 0.1940 μM, respectively (Fig. 4e, f).

**ASS1 functions as a tumor suppressor in BC.** ASS1 is essential for the removal of waste nitrogen by catalyzing the condensation of citrulline and aspartate to form argininosuccinate. Genetic ASS1 deficiency causes a urea cycle disorder citrullinemia. The somatic silence of ASS1 is very common in various types of cancers, however, the role of ASS1 in cancer is unclear[6,7,26].

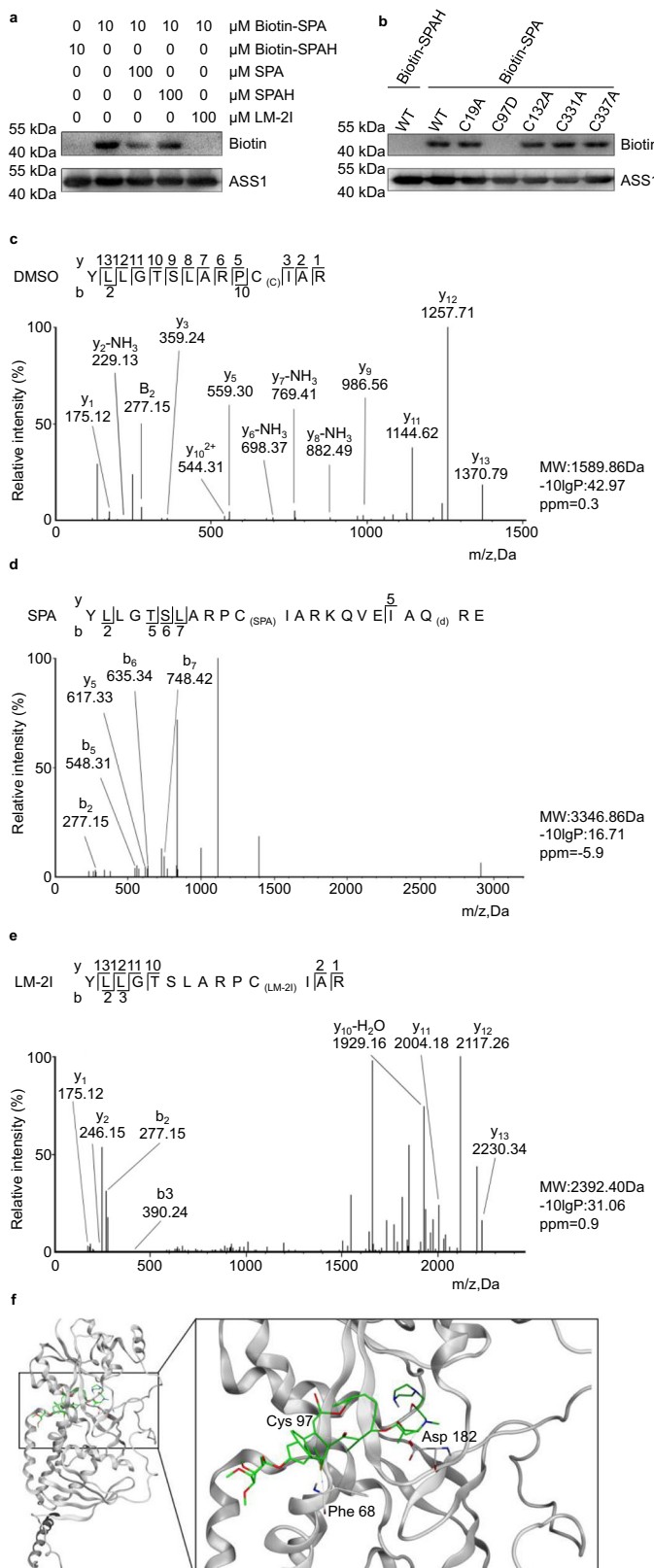

**Fig. 3 SPA and LM-2I target ASS1 at C97 site. a** The recombinant ASS1 protein was incubated with Biotin-SPA in the absence or presence of SPA, SPAH and LM-2I for 1 h at room temperature, followed by western blot using streptavidin-HRP or ASS1 antibody (*n* = 3 biologically independent experiments). **b** Recombinant WT and mutant ASS1 proteins were incubated with Biotin-SPA or Biotin-SPAH for 30 min, followed by western blot using streptavidin-HRP or ASS1 antibody (*n* = 3 biologically independent experiments). MS/MS analysis for the C97-containing peptide of recombinant ASS1 pre-incubated with vehicle (**c**), SPA (**d**), or LM-2I (**e**). (SPA), combined with SPA; (LM-2I), combined with LM-2I; (C), carbamidomethylation; (d), deamidation. **f** Binding model of LM-2I-ASS1. The left panel is the global view of the entire ASS1 structure, the right panel is the focused view of LM-2I in the binding site (ASS1 in gray while LM-2I in green). Source data are provided as a Source Data file.

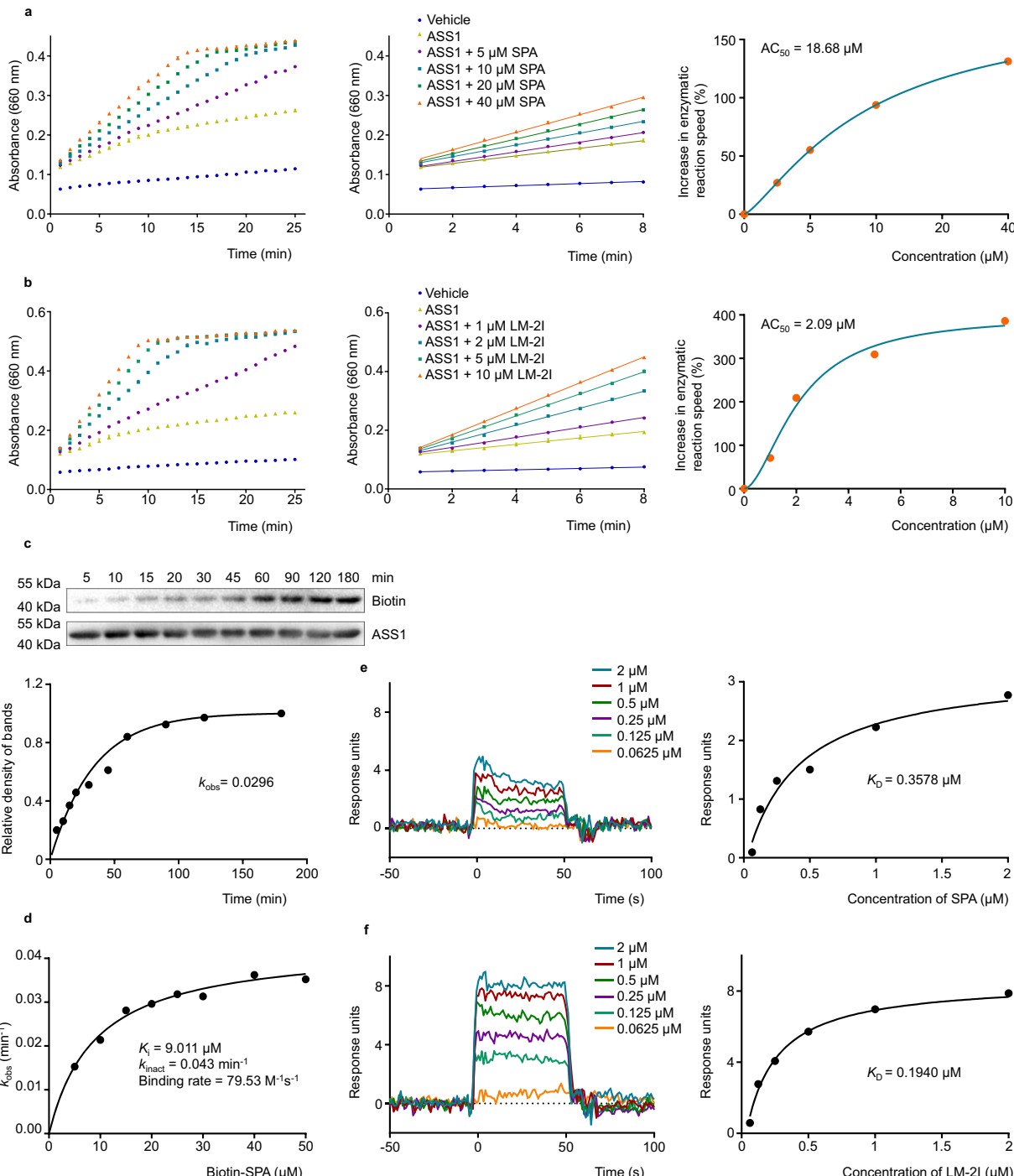

**Fig. 4 SPA and LM-2I enhance ASS1 catalytic activity and its kinetics.** The recombinant ASS1 protein was incubated with different concentrations of SPA (**a**) or LM-2I (**b**) overnight, and the ASS1 catalytic activity was measured. The initial increase in absorbance at 660 nm ($A_{660}$) between 1 and 8 min is expanded in the middle panel. In the right panel, the Hill 1 model of Origin 2018 software was used to analyze the increase in enzymatic reaction speed at different concentrations. The maximal activation ratio and $AC_{50}$ were calculated. (mean ± s.d.). **c** The determination of $k_{obs}$ for the interaction of ASS1 (0.5 μM) with Biotin-SPA (20 μM) was calculated at different time points ($n = 3$ biologically independent experiments). **d** Plotting the $k_{obs}$ values for binding ASS1 (0.5 μM) with different concentrations of Biotin-SPA. BIACore diagram (left panel) and $K_D$ (right panel) of the recombinant ASS1 protein binding to SPA (**e**) or LM-2I (**f**). Source data are provided as a Source Data file.

To define the role of ASS1 in breast cancer, we examined the expression of ASS1 (Fig. 5a) in a series of breast cancer cell lines, and found that MCF-7 cells have relative high while MDA-MB-231cells have relative low levels of ASS1 expression (Fig. 5a). MDA-MB-231cells were then used to generate stable ASS1 overexpression cells by transfecting ASS1 expression plasmid (Fig. 5b, top) while MCF-7 cells were employed to establish stable ASS1 knockdown cells by infecting ASS1 shRNA lentivirus (Fig. 5c, top). Consistently, overexpression of ASS1 in MDA-MB-231 cells decreased (Fig. 5b, bottom) while knockdown of ASS1 in

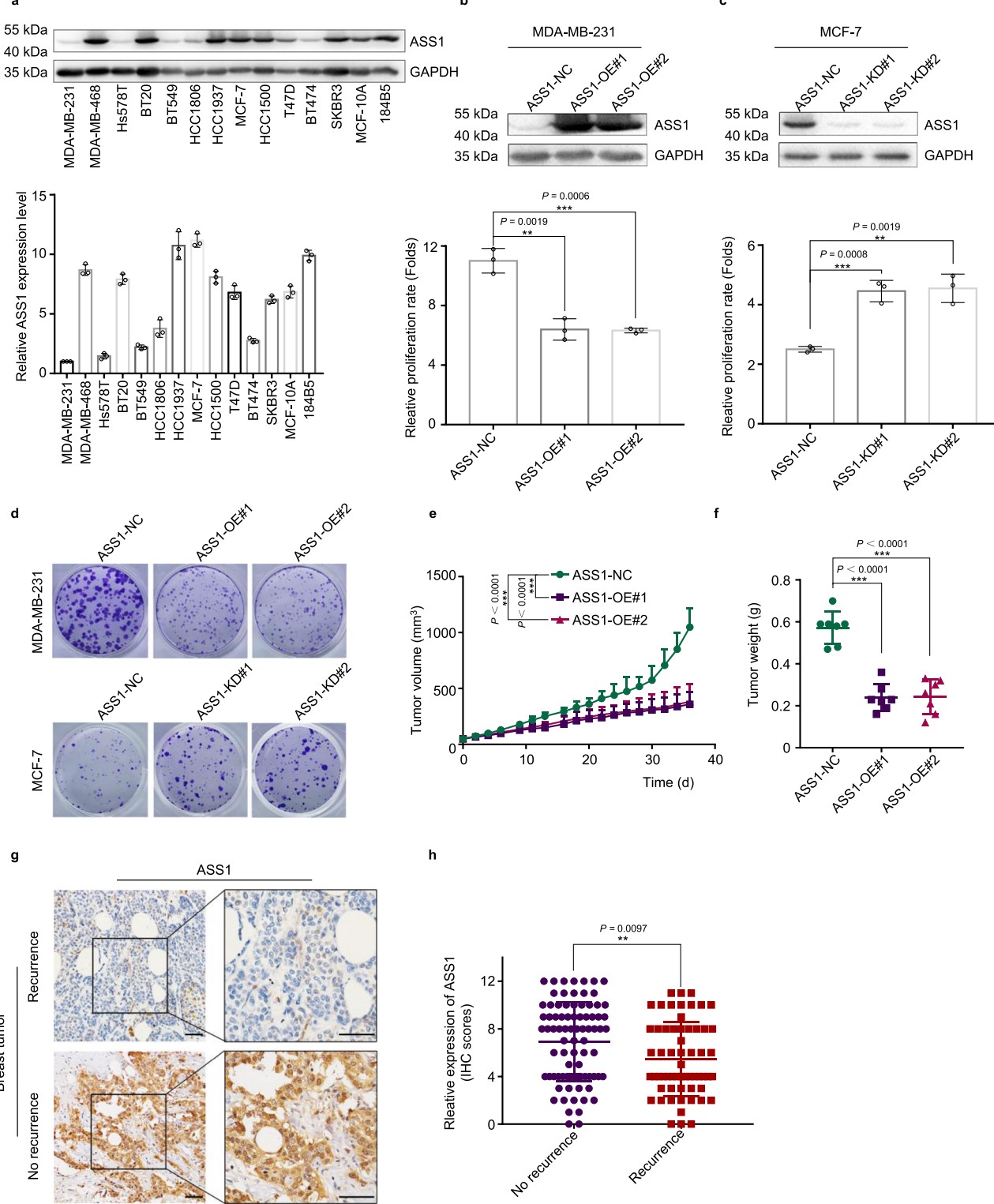

MCF-7 cells increased the cell proliferation rate (Fig. 5c, bottom) and colony formation (Fig. 5d). Importantly, overexpression of ASS1 in MDA-MB-231 cells significantly inhibited xenograft tumor development in mouse models (Fig. 5e and f). Furthermore, we detected and analyzed the expression of ASS1 in clinical human breast cancer tissue samples by immunohistochemistry (Fig. 5g), we found that the down-regulation of ASS1 expression in human breast cancer was significantly correlated with tumor recurrence within three years after surgery and adjuvant therapy

(Fig. 5h). Altogether, these results support that ASS1 act as tumor suppressor in breast cancer.

**SPA and LM-2I function as tumor inhibitor via directly targeting ASS1.** Our results showed that the C13-C14 double bond in SPA and LM-2I (Fig. 3a) was essential for the binding of ASS1 to SPA and LM-2I, and the tumor inhibitory effect of SPAH was significantly reduced as compared with SPA (Fig. 2b and

**Fig. 5 ASS1 functions as a breast tumor suppressor. a** Western blot for ASS1 expression in breast cancer cell lines (top). The relative quantification of ASS1 expression in breast cancer cell lines (ASS1 expression in MDA-MB-231 cells was defined as 1) (bottom) (mean ± s.d., $n = 3$ biologically independent experiments). Western blot for the expression of indicated proteins in MDA-MB-231 cells transfected with *ASS1* expression plasmid (ASS1-OE) or empty vector (**b**, top), and in MCF7 cells infected with control or *ASS1* shRNA lentivirus (ASS1-KD) (**c**, top). Crystal violet assay for the proliferation rate of MDA-MB-231 cells transfected with *ASS1* expression plasmid (ASS1-OE) or empty vector (b, bottom), and for MCF7 cells infected with control or *ASS1* shRNA lentivirus (ASS1-KD) (c, bottom) (mean ± s.d., $n = 3$ biologically independent experiments). **$P < 0.01$, ***$P < 0.001$ from two-tailed unpaired Student's t-tests.**d** Colony formation of MDA-MB-231 cells transfected with *ASS1* expression plasmid (ASS1-OE) or empty vector, and of MCF7 cells infected with control or *ASS1* shRNA lentivirus (ASS1-KD). $1 \times 10^6$ MDA-MB-231 cells stably expressing ASS1 (ASS1-OE) or control vector (NC) were injected subcutaneously into the right flank of BALB/c nu/nu female mice. Tumor sizes were measured every 2 or 3 days (**e**). 36 days later mice were sacrificed, tumors were dissected and weighted (**f**). Data represented as mean ± s.d. ($n = 7$ mice per group). ***$P < 0.001$ from two-tailed unpaired Student's t-tests. **g** Representative imagines of ASS1 immunohistochemical staining in paraffin-embedded human breast cancer tissue sections. Scale bars are 50 μm. **h** Comparing the expression levels of ASS1 in recurrent ($n = 60$) and no recurrent ($n = 83$) human breast cancers. **$P < 0.01$ from two-tailed unpaired Student's t-tests. Source data are provided as a Source Data file.

Supplementary Fig. S6a, b). We further found that SPA or LM-2I pre-incubated with ASS1 recombinant protein had significantly less toxicity to cancer cells than the same concentration of SPA (Supplementary Fig. S7a) or LM-2I (Fig. 6a). These results indicate that the physical binding of SPA and LM-2I to ASS1 is essential for SPA and LM-2I -induced tumor inhibition.

To confirm that SPA and LM-2I -induced tumor inhibition is indeed via targeting ASS1, we established ASS1 stable knockout (ASS1-KO) MDA-MB-231 cells by using CRISPR/Cas9 system. ASS1-KO and control MDA-MB-231 cells were treated with different concentrations of SPA or LM-2I, and the cell viability was measured by crystal violet assay. We found that ASS1-KO MDA-MB-231 cells did not respond or were very insensitive to the treatment of SPA (Supplementary Fig. S7b) and LM-2I (Fig. 6b), suggesting that the anticancer activity of SPA and LM2I is indeed through targeting ASS1.

Interestingly, we found that in MCF-7 cells, which has high level of endogenous ASS1 expression, knockdown of ASS1 sensitized MCF-7 cells to SPA (Supplementary Fig. S7c) and LM-2I (Fig. 6c) inhibition, while in MDA-MB-231, which has much low level of endogenous ASS1 expression, over-expression of exogenous ASS1 significantly decreased the sensitivity of MDA-MB-231 cells to the inhibition of SPA (Supplementary Fig. S7d) and LM-2I (Fig. 6d). Consistently, the inhibitory effect (IC$_{50}$) of SPA (Supplementary Fig. S7e) and LM-2I (Fig. 6e) on the proliferation of different breast cancer cell lines was significantly correlated with the endogenous expression levels of ASS1 in the cells (Figs. 1b, e, and 5a). These results indicate that cancer cells with lower level of ASS1 expression are more sensitive to SPA and LM-2I.

It has been recently reported that decreased activity of ASS1 in cancers supports proliferation of cancer cells by facilitating pyrimidine synthesis[6]. To investigate whether SPA and LM-2I inhibit pyrimidine synthesis, we examined the metabolites of MDA-MB-231 cells treated with SPA, LM-2I or vehicles for 24 h by LC-MS analysis. We found that SPA or LM-2I treatment significantly reduced the metabolites, such as aspartic acid, CMP, UMP and TMP, in MDA-MB-231, MCF-7 and MCF-10A cells (Fig. 6f and Supplementary Fig. S8a, b). Importantly, SPA and LM-2I -induced death of MDA-MB-231, MCF-7 and MCF-10A cells could be rescued by addition of pyrimidines in the medium, however, the addition of purines had no such effect (Fig. 6g and Supplementary Fig. S8c, d). These results further confirm that the antitumor activity of SPA and LM-2I is via promoting ASS1 activity, leading to reduced pyrimidine synthesis and the inhibition of cell proliferation.

## Discussion

The expression of ASS1 is increased in some types of tumors, such as colorectal, ovarian, stomach, and lung cancers, while in

most types of cancers, such as melanoma, prostate cancer, breast cancer, bladder cancer, hepatocellular carcinoma, mesothelioma, pancreatic cancer, nasopharyngeal carcinoma, osteosarcomas, and myxofibrosarcomas, the expression of ASS1 is decreased[3–6]. Similarly, some reports suggest that ASS1 plays a oncogenic role in colorectal cancer[26] while other reports support that ASS1 has tumor suppressor role in many other types of tumors[6,8]. Our study demonstrates that ASS1 knockdown increases while ASS1 overexpression decreases breast cancer cell proliferation and tumor development, and that down-regulation of ASS1 expression is significantly related to tumor relapse, indicating that ASS1 function as tumor suppressor in breast cancer. Consistent with our studies, low ASS1 expression in tumor tissues is associated with poor prognosis of breast cancer and hepatocellular carcinoma patients[7,8]. Therefore, it seems that the discrepancy of the role of ASS1 is related to its expression level in cancers. Our studies and the data in literature[6–8] support that ASS1 functions as a tumor suppressor in those types of cancers whose ASS1 expression is down-regulated.

Tumors with silence or down regulation of ASS1 are unable to synthesize arginine and are therefore auxotrophic or depending on external supplementation of arginine for the growth and survival[6,9,10]. Many efforts have been made to exploit this metabolic vulnerability and led to the development of arginine deprivation therapies (ADT)[6–8,12,13]. Despite promising results observed in preclinical studies and clinical phase I–III trials, some tumors are resistant to ADT[12,27]. Theoretically, based on the principle of the strategy, ADT is more effective in ASS1-negative tumors than in those tumors in which ASS1 is still expressed although at low level. And actually it has been reported that the regain of ASS1 expression is one of the mechanisms by which tumors become resistant to ADT[12]. In the present study, we have demonstrated that SPA and its derivatives significantly inhibit cancer cell proliferation and tumor growth by activation of ASS1 in tumors. Treatment with SPA and LM-2I results in significant enhancement of ASS1 enzymatic activity in cancer cells, particularly in those cancer cells with low ASS1 expression, leading to the inhibition of cancer cell proliferation. Therefore, our studies lay the foundation for the development of novel therapeutic strategies for tumors with low ASS1 expression.

SPA is one of the two components of the insecticide spinosad, a natural substance produced by *Saccharopolyspora spinoza*[17]. Because of its low mammalian toxicity, spinosad has been widely used to control a wide variety of pests[17–19]. Spinosad interrupts the function of nicotinic and GABA-gated ion channels, resulting in rapid excitation of the insect nervous system, leading to involuntary muscle contractions, tremors, paralysis, and death[28]. Our studies demonstrate that SPA and its derivatives are potent tumor inhibitors. Consistent to their low toxicity to mammals, SPA and its derivative LM-2I show very mild toxicity to normal

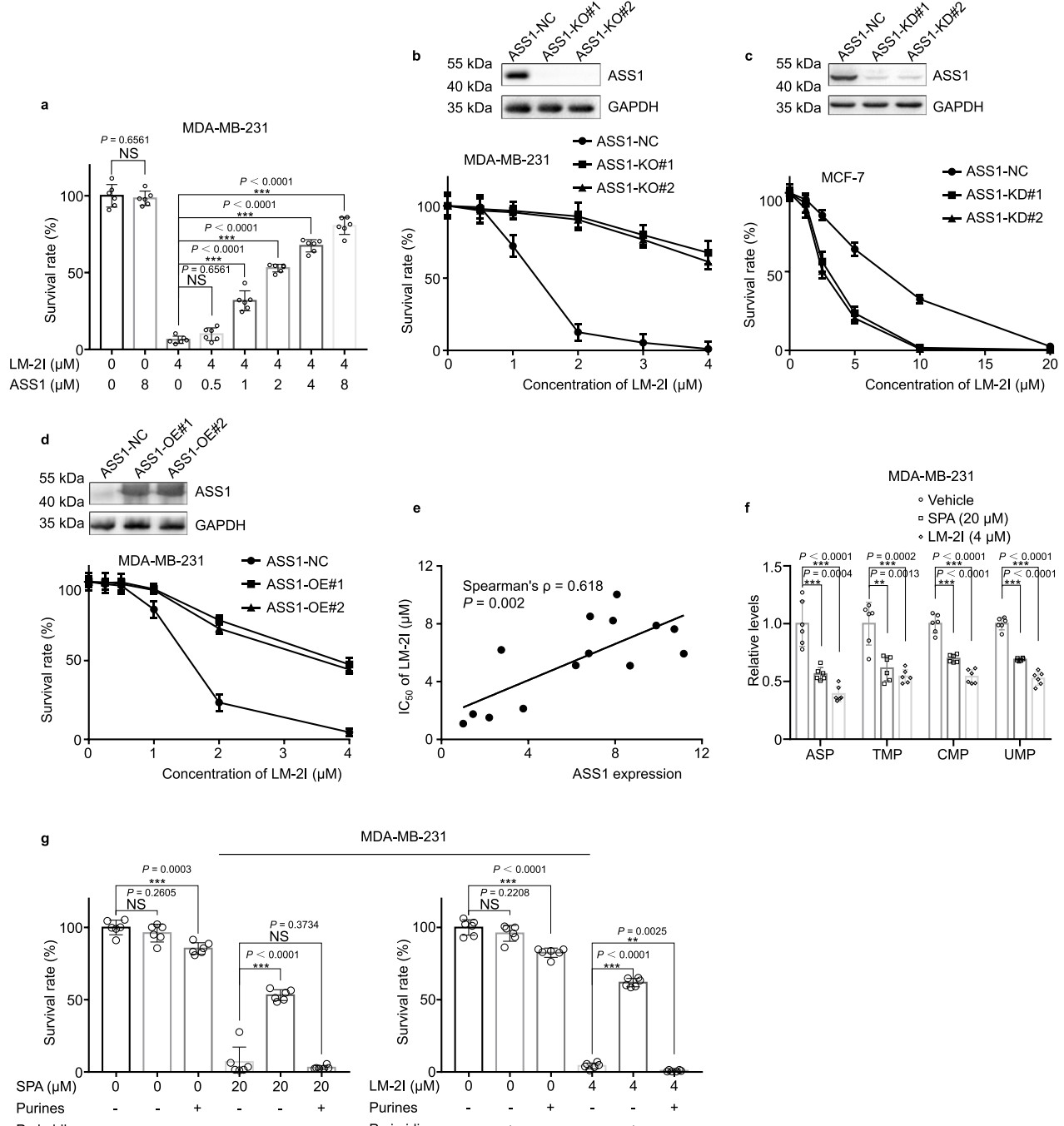

**Fig. 6 ASS1 is indeed the target of SPA and LM-2I. a** Crystal violet assay for the viability of MDA-MB-231 cells treated with LM-2I or LM-2I pre-incubated with different concentrations of ASS1 recombinant protein (mean ± s.d., $n = 6$ biologically independent experiments). \*\*\*$P < 0.001$, NS: not significant ($P > 0.05$) from two-tailed unpaired Student's $t$-tests. **b** Western blot for ASS1 expression in control or stable ASS1 knockout (ASS1-KO) MDA-MB-231 cells (top). Crystal violet assay for the viability of control and ASS1-KO MDA-MB-231 cells treated with LM-2I (bottom) (mean ± s.d., $n = 3$ biologically independent experiments). **c** Western blot for ASS1 expression in control or stable ASS1 knockdown (ASS1-KD) MCF-7 cells (top). Crystal violet assay for the viability of control and ASS1-KD MCF-7 cells treated with LM-2I (bottom) (mean ± s.d., $n = 3$ biologically independent experiments). **d** Western blot for ASS1 expression in control or stable ASS1 overexpression (ASS1-OE) MDA-MB-231 cells (top). Crystal violet assay for the viability of control and ASS1-OE MDA-MB-231 cells treated with LM-2I (bottom) (mean ± s.d., $n = 3$ biologically independent experiments). **e** The correlation between the relative ASS1 expression in cells and the $IC_{50}$ of these cells treated with LM-2I ($n = 3$ independent experiments from $n = 14$ cell lines). $P$ values of mean ± s.d. were determined by Pearson Coefficient. **f** LC-MS analysis of ASP, UMP, CMP, and TMP in MDA-MB-231 cells treated with SPA, LM-2I or vehicles for 24 h (mean ± s.d., $n = 6$ biologically independent experiments). \*\*$P < 0.01$, \*\*\*$P < 0.001$ from two-tailed unpaired Student's $t$-tests. **g** Crystal violet assay for the viability of MDA-MB-231 cells preincubated with pyrimidines or purines, followed by treatment with SPA or LM-2I (mean ± s.d., $n = 6$ biologically independent experiments). \*\* $P < 0.01$, \*\*\*$P < 0.001$, NS: not significant ($P > 0.05$) from two-tailed unpaired Student's $t$-tests. Source data are provided as a Source Data file.

cells and mice although they strongly inhibit cancer cell proliferation in vitro and tumor growth in mouse models, suggesting that SPA and its derivatives are very good candidates for clinical cancer drug development.

Epigenetic silencing or down-regulation of ASS1 expression is one of the major mechanisms contributing to the loss of the tumor suppressor role of ASS1 in cancers[12]. In some tumors, such as glioblastoma, bladder cancer and hepatocellular carcinoma, methylation of the promoter region of the *ASS1* gene mediates its silencing[4]. The expression of ASS1 in tumors cells can be changed in response to some cancer therapies. For instance, ASS1 is directly transactivated by p53 in response to genotoxic stress[29], while under the conditions of arginine deiminase-based arginine depletion the expression of ASS1 in melanoma cells is controlled by the interplay among transcriptional regulators c-Myc, Sp4, and HIF-1alpha[30]. Modification of ASS1 protein is another important factor that leads to loss of its enzymatic activity and tumor suppressor role. It has been reported that the protein CLOCK (circadian locomotor output cycles kaput) directly acetylates ASS1 at K165 and K176 sites to inactivate ASS1 in circadian rhythms[31]. Nitric oxide mediates reversible S-nitrosylation of ASS1 at Cys132 site to inactivate ASS1 in vitro and in vivo[32]. Our studies demonstrate that SPA and LM-2I act as ASS1 agonists, which interact with and activate ASS1, leading to enhanced antitumor activity of ASS1. The Cys97 (C97) site of ASS1 is critical for its activation by SPA and LM-2I.

ASS1 is a key enzyme in de novo arginine synthesis pathway, any dysfunction of ASS1 will lead to the rearrangements of arginine metabolism[13,29,33]. For instance, ASS1-deficient malignant mesothelioma cells have decreased levels of acetylated polyamine metabolites and a compensatory increase in the expression of polyamine biosynthetic enzymes[13]. It has been recently reported that decreased activity of ASS1 promotes cell proliferation by facilitating pyrimidine synthesis through activation of an enzyme complex composed of carbamoyl-phosphate synthase 2, aspartate transcarbamylase, and dihydroorotase, while blocking pyrimidine synthesis reduces cell proliferation[6]. Our studies demonstrate that SPA or LM-2I treatment significantly reduces the metabolites, such as urea, aspartic acid, CMP, UMP and TMP, while the addition of pyrimidines can rescue SPA and LM-2I -induced cancer cell death, suggesting that the antitumor activity of SPA and LM-2I is indeed through activation of ASS1, which results in reduced pyrimidine synthesis and consequently the inhibition of cancer cell proliferation.

Since SPA and LM-2I function as potent ASS1 agonist in cancer cells with low ASS1 expression, it would be very interesting to test whether they also act as ASS1 activator in those nontumor diseases caused by ASS1 deficiency or down-regulation. As some of these non-tumor diseases, such as citrullinemia type I (CTLN1), are caused by mutations in ASS1 gene or enzyme-defected splicing variants[34–36]. Therefore, it would be very important to investigate whether SPA and LM-2I can activate or de-inactivate the mutant ASS1 proteins or/and the enzyme-defected splicing variants identified in CTLN1.

## Methods

**Antibody information.** Anti-ASS1 (#70720), Anti-Ki-67 (#9449), anti-GAPDH (#2118), anti-rabbit IgG-HRP (#7074), and streptavidin-HRP (#3999) were purchased from Cell Signaling Technology (CST), Streptavidin-FITC (11-4317-87) were from eBioscience, and goat anti rabbit IgG-CY3 (111-167-003) were from Jackson ImmunoResearch.

**Cell culture.** All cell lines were purchased from American Type Culture Collection (ATCC, Manassas, VA, USA). The human breast cancer (BC) cell lines BT549, BT474 and T47D were maintained in RPMI 1640 containing 10% (v/v) fetal bovine serum (FBS, Gibco), 0.01 mg/mL insulin and 100 units/mL penicillin/streptomycin (HyClone). SKBR3, Hs578T, and BT20 human BC cell lines were cultured in

DMEM medium with 10% (v/v) FBS and 100 units/mL penicillin/streptomycin. HCC1500, HCC1806, and HCC1937 human BC cell lines were cultured in RPMI 1640 medium supplement with 10% (v/v) FBS and 100 units/mL penicillin/streptomycin. MDA-MB-231 human BC cell line was cultured in DMEM/F12 (1: 1) medium with 10% (v/v) FBS and 100 units/mL penicillin/streptomycin. MDA-MB-468 human BC cell line was cultured in DMEM (high glucose) medium with 10% (v/v) FBS and 100 units/mL penicillin/streptomycin. MCF-7 human BC cell line was cultured in EMEM medium with 10% (v/v) FBS, 0.01 mg/mL insulin and 100 units/mL penicillin/streptomycin. The human immortalized breast epithelial cell lines MCF10A and 184B5 were maintained in DMEM/F12 (1:1) medium with 5% (v/v) horse serum, 500 ng/mL hydrocortisone, 0.01 mg/mL insulin, 20 ng/mL epidermal growth factor, 100 ng/mL cholera toxin and 100 units/mL penicillin/streptomycin. All of the cells cultured in a humidified incubator at 37 °C and 5% $CO_2$/95% air (v/v).

**Biotin-SPA-binding proteins pull-down and MS analysis.** MCF-7 cell lysate was divided into four equal volumes, in which the drug competition groups were incubated with different concentrations of LM-2I at room temperature for 4 h, and then incubated with biotin or Biotin-SPA for overnight at 4 °C, followed by pull down with streptavidin Magnetic beads (Life Technologies). The proteins binding to streptavidin Magnetic beads were separated by SDS-PAGE and silver staining. The specific bands were cut and digested in-gel for LC-MS/MS analysis.

**Preparation of recombinant wild-type and mutant ASS1 proteins.** Human *ASS1* gene was cloned into the pET28a vector with a 6×His tag at N-terminal. The mutant *ASS1* gene was generated by site directed mutagenesis using Mut Express II Fast Mutagenesis Kit V2 (Vazyme) in pET28a-*ASS1* vector. The BL21 (DE3) E. coli were transformed by wild-type or mutant *ASS1* expression plasmids, and the recombinant proteins were purified by Ni-NTA (QIAGEN).

The CDS (the coding sequence) of human *ASS1* variant 1 (ACCESSION: NM_000050) was used for construction of prokaryotic expression plasmids.

Primer sequences are shown in Supplementary Table S2.

**ASS1 knockdown and overexpression.** Control and lentiviral vectors expressing ASS1 or ASS1 shRNA, purchased from GenePharma, were transfected into HEK293T cells using RNAi-Mate to produce control, ASS1 overexpression, or ASS1 knockdown lentiviruses. To establish ASS1 overexpression or knockdown cells, MDA-MB-231 cells were infected with control or lentiviruses expressing ASS1 or ASS1 shRNA, and selected with 5 μg/mL puromycin. The short hairpin sequences used for generation of the ASS1 knockdown cell lines were:

*ASS1* shRNA1:
TGGATGTCAGCAGGGAGTTTGTTTCAAGAGAACAAACTCCCTGCTGA
CATCCTTTTTTC.

*ASS1* shRNA2:
TGGAGGATGCCTGAATTCTACATTCAAGAGATGTAGAATTCAGGCAT
CCTCCTTTTTTC.

The negative control shRNA sequence:
TGTTCTCCGAACGTGTCACGTTTCAAGAGAACGTGACACGTTCGGAG
AACTTTTTTC.

The CDS (the coding sequence) of human ASS1 variant 1 (ACCESSION: NM_000050) was used for construction of eukaryotic expression plasmids.

**Establishment of ASS1 KO cell lines using the CRISPR/Cas9 system.** ASS1 guide RNAs (gRNAs) were designed using the online CRISPR design tool (http://crispor.tefor.net/). The construction of pLentiCRISPR was performed according to the protocol provided by the Zhang Lab (http://genome-engineering.org/gecko/). Oligos gRNAs were cloned into plentiCRISPR v2 and packaged into lentiviruses. To obtain stable ASS1 KO cells, cells were transfected with the pLentiCRISPR v2 plasmid containing each target gRNA sequence or empty vector, selected with puromycin for 3 days and isolated by colony formation assay. The single clones were validated by immunoblotting analysis and DNA sequencing. Primer sequences are shown in Supplementary Table S2.

**Crystal violet assay.** Cells were seeded in 96-well plates at 2000–5000 cells per well.12 h later, drugs were added and incubated for 48 h, then cells were washed with PBS X1 and fixed in 4% PFA (in PBS). Cells were then stained with 0.1% Crystal Violet (C0775, Sigma-Aldrich) for 20 min (1 ml per well) and washed with water. Cells were then incubated with 10% acetic acid for 20 min with shaking. The absorbance of extract was measured at 590 nm.

**Immunofluorescence assay.** Cells grown on slides were treated with different conditions and then fixed with 4% paraformaldehyde. Cells were treated with 0.3% Triton-X100 and 3% BSA in PBS, and then incubated with primary antibodies for 3 h, followed by incubated with the secondary antibodies for 1 h at room temperature. All slides were counterstained with DAPI, and examined under a Leica TCS SP8 X&MP laser scanning confocal microscope (LASX V3.5.2.18963).

**Colony formation assays**. Cells were seeded in 12-well plates (600 cells/well for MDA-MB-231; 1500 cells/well for MCF-7; 500 cells/well for MCF-10A). The cells were treated with SPA, LM-2I, or vehicles, and the medium was replaced every 3 days until visible colonies were formed. At the end of the experiment, cells were fixed with 4% paraformaldehyde for 30 min and stained with 0.1% crystal violet in 20% ethanol for 30 min.

**Western blot analysis**. Cell lysates were separated by SDS-PAGE and transferred to PVDF membranes. Immunostaining was done using specific primary antibodies. Chemiluminescence detection was performed by using the Pierce ECL Western Blotting Substrate (Thermo Scientific). The fluorescence signals were collected using Image Lab software (v4.1), and the quantification of western blots were analyzed using Image J version 1.48.

**Mouse xenograft tumor models**. All animal experiments were performed in accordance with China FDA guidelines. Protocols were reviewed and approved by the Department of Laboratory Animals, Central South University. Four to six weeks old BALB/c nu/nu female mice were purchased from SJA Laboratory Animal Co., Ltd (Hunan). $5 \times 10^6$ MDA-MB-231 cells were inoculated subcutaneously into the right flank of mice. When the average tumor size reached about 100 mm$^3$, the mice were randomly divided into 3 groups and treated with vehicle, SPA (10 mg/kg/day), or LM-2I (5 mg/kg/day), i.p. every other day. Tumor size and mouse body weight were measured every two or three days. When the tumor size in the vehicle-treated group exceeded 1000 mm$^3$ (28 days after treatment), mice were sacrificed. the tumors were dissected and weighed. Tumor volume was calculated using the equation $V$ (mm$^3$) $= (a \times b^2)/2$, where a is the largest diameter and b is the smallest diameter.

**Determination of the interaction kinetics between ASS1 and SPA or LM-2I**. The covalent binding between ASS1 and Biotin-SPA (SPA or LM-2I) is a two-step reaction. The first step is the non-covalent reversible binding (ASS1:Biotin-SPA) followed by the formation of irreversible covalent bonds (ASS1-Biotin-SPA). $K_i$ represents the dissociation constant of the ASS1:Biotin-SPA complex, while $k_{inact}$ is a rate constant for ASS1-Biotin-SPA formation at Biotin-SPA saturating. To calculate the $K_i$ and $k_{inact}$ values, we incubated ASS1 with an excess of Biotin-SPA for a different period of time and then streptavidin-HRP was used to detected Biotin-SPA by western blot. The optimal exposure time was adjusted to ensure that the strip grayscale was proportional to the Biotin-SPA concentration, and finally the amount of Biotin-SPA covalently bound to ASS1 was determined by quantitative grayscale scanning. Under the psendo-first order experimental conditions, the reaction follows as the equations:

$$[ASS1]_t = [ASS1]_0 \times E^{-kobs \times t} \tag{1}$$

$$[ASS1 - Biotin - SPA] = [ASS1]_0 - [ASS1]_t = [ASS1]_0 \times (1 - E^{-kobs \times t}) \tag{2}$$

$[ASS1]_0$ represents the total concentration of ASS1 in the reaction system, and $[ASS1]_t$ is the concentration of ASS1 at time t. The $k_{obs}$ for different concentrations of Biotin-SPA is calculated by equation (2). The value of $k_{inact}$ and $K_i$ were obtained by fitting the value of $k_{obs}$ into equation (3).

$$k_{obs} = k_{inact} \times [Biotin - SPA]/([Biotin - SPA] + K_i) \tag{3}$$

**Molecular modeling**. ASS1 protein was used as the docking template (www.wwpdb.org (PDB code: 2NZ2). The QuickPrep module of Molecular Operating Environment (MOE) 2018 is used to process the protein, including hydrogenation, charge, water removal and energy minimization. Meanwhile, the WASH module of MOE was used for hydrogenation and energy minimization of small molecules. The covalent dock module was used to manually dock proteins and small molecules, and Michael acceptor was used as the functional group for 1,4-addition. Other parameters were kept at default.

**ASS1 activity assay**. Recombinant ASS1 was incubated with different concentrations of SPA or its derivatives at 4 °C for overnight, and then the reaction buffer [20 mM Tris. HCl (pH 7.8), 2 mM ATP, 10 mM citrulline, 10 mM aspartate, 6 mM MgCl$_2$, 20 mM KCl, and 0.2 U pyrophosphatase (l1643, Sigma)] was added. The reaction was incubated for different periods of time at 37 °C, followed by adding 50 μL molybdate buffer [10 mM ascorbic acid, 2.5 mM ammonium molybdate, 2% (v/v) sulfuric acid]. The absorbance at 660 nm was collected using EnSpire software (v 4.1). Increased absorbance corresponding to increased ASS1 activity was determined by pyrophosphate production.

For calculation, select the linear time period of the enzymatic reaction, and use the spss18 software to fit the rate (i.e., slope) of the enzyme reaction at each compound concentration. Subsequently, $V_0$ represents the rate of no ASS1, $V_{vehicle}$ represents the rate of ASS1 + Vehicle, and $V_{ASS1 + XμM}$ represents the rate at each drug concentration. Then, the percentage of activation of ASS1 with different concentrations of drugs $= (V_{A + X μM} - V_{vehicle}) \times 100\%/(V_{vehicle} - V_0)$. Finally, use the Hill1 model of the Origin 2018 software to fit and calculate the maximal activation ratio and AC$_{50}$.

**Surface plasmon resonance (SPR)**. The SPR analysis was performed using a BIAcore X100 (GE healthcare) at 25 °C. The purified ASS1 was covalently coupled onto the CM5 sensor chip. SPA and LM-2I were dissolved in HBS-EP buffer (GE healthcare). After filtered by 0.22 μM filter, the binding reaction was performed to obtain the sensing curve with different concentrations. BIAevaluation software (v4.1) was used to collect data and analyze. According to the characteristics of fast binding and fast disassociation of the curves, the state affinity curve fit model was used to fit $K_D$ value.

**Human subjects**. Breast cancer tissues were obtained from the Second Xiangya Hospital of Central South University. Informed consent was obtained from all participants in accordance with the Declaration of Helsinki. All protocols using human specimens were approved by the institutional Review Board of the Second Xiangya Hospital of Central South University. The clinical features of the patients are listed in Supplementary Table S1.

**Immunohistochemistry staining (IHC)**. IHC staining was performed using 4 μm paraffin-embedded tumor tissue sections. The primary antibodies against ASS1 (70720, Cell Signaling Technology, USA) and Ki-67 (9449, Cell Signaling Technology, USA), were diluted 1:250 or 1:500, respectively, and then incubated at 4 °C overnight in a humidified container. After three washes with PBS, the tissue slides were treated with a nonbiotin horseradish peroxidase detection system according to manufacturer's instructions (ZSGB-BIO, China) and photographed using an LEICA DM3000 LED (Leica DMshare (v3), Germany). IHC staining was evaluated by two independent pathologists. The Ki67 and ASS1 signals were detected in the nucleus and cytoplasm, respectively. To evaluate Ki67 and ASS1, a semiquantitative scoring criterion was used in which both the staining intensity and positive areas were recorded. A staining index (values 0–12), which was obtained as the product of the intensity of positive staining (weak, 1; moderate, 2; strong, 3) and the proportion of positive cells of interest (0%, 0;<25%, 1; 26–50%, 2; 51–75%, 3; >76%, 4), was calculated.

**Hematoxylin and eosin (HE) staining**. HE staining was conducted according to routine protocols. Briefly, after deparaffinization and rehydration, tissue sections were stained with hematoxylin solution (ZSGB-BIO, China) for 5 min followed by 5 dips in 1% acid ethanol (1% HCl in 75% ethanol) and then rinsed in distilled water. Then the sections were stained with eosin solution (ZSGB-BIO, China) for 3 min and followed by dehydration with graded alcohol and clearing in xylene. The mounted slides were then examined and photographed using an LEICA DM3000 LED (Leica DMshare (v3), Germany).

**Metabolomics analysis**. MCF-7, MDA-MB-231, and MCF-10A cell lines were seeded at $1.5 \times 10^6$ to $2 \times 10^6$ cells per 10 cm dish overnight, and then cells were treated with SPA or LM-2I for 24 h. Subsequently, the cells were washed with ice-cold PBS, lysed with 500 μL of metabolite extract (methanol/acetonitrile/water = 4/4/2) and quickly scraped. Following mixing and sonication, the supernatant was isolated, lyophilized, and stored at −80 °C. Frozen samples was re-suspended in 500 μL 5 M ammonium formate/acetonitrile (99.1/0.9), and injected in a volume of 1 μL.

The LC-MS/MS instrument was an AB SCIEX QTRAP® 6500 + LC–MS/MS system equipped with an electrospray ion source and operated in an anion mode for the analysis of nucleoside monophosphates. Analyst (version 1.6.3, AB) and MultiQuant software (version 3.0.2, AB) were applied for data acquisition and analysis. The chromatographic separation was carried out on a 100 mm × 2.1 mm inner diameter, 1.7 μm C18 100 Å UPLC Kinetex® column, mobile phase at 5 M ammonium formate/acetonitrile = 99.1/0.9, flow rate at 0.2 mL/min, column temperature at 40 °C. The sample kept at 8 °C was automatically injected into a volume of 1 μL.

For mass spectrometry, nitrogen was used as the collision gas, the curtain gas was 35.0 psi, and the collision gas was medium. The electronic spray voltage was set to −4.5 kV, temperature 450 °C, spray gas 55.0 psi, collision voltage −16.000 V, de-clustered voltage −60.000 V, inlet voltage −10.000 V, collision cell exit voltage −1.500 V, dwell time 130.0 ns.

**Chemical synthesis**. All chemicals were of analytical grade and used as received unless otherwise stated. $^1$H and $^{13}$C NMR spectra were recorded on a Bruker DRX 400 or Bruker DRX 500 NMR instrument. Chemical shifts are given in ppm, and coupling constants are given in hertz. MS were recorded on an Agilent 6200/6500 series Q-TOF MS (HR-ESI-MS) system. Melting points were determined on an X-4 Digital micro-melting point apparatus without correction. Spinosyn A was obtained by dissolving commercially available spinosad in methanol, followed by filtration and evaporation under reduced pressure.

*N*-2-hydroxy-3-(piperazin-1-yl)propyl spinosyn B (LM-2I, **2**) : To a solution of *N*-3-chloro-2-hydroxypropyl spinosyn B (100 mg, 0.123 mmol) in methanol (10 mL), piperazine (135 mg, 1.23 mmol) was added. The reaction mixture was stirred for 48 h at 50 °C. After evaporation of organic solvent under reduced pressure, the resulting residue was added water (50 mL) and then extracted with ethyl acetate (3 × 25 mL). The organic layer was collected, dried over anhydrous sodium sulfate, and removed under reduced pressure. The resulting crude product

was purified by silica gel column chromatography using a mixed solvent of ethyl acetate, methanl and triethylamine (5: 1: 0.05, v/v) as an eluent affording *N*-2-hydroxy-3-(piperazin-1-yl)propyl spinosyn B (**2**) as a white glassy solid (72 mg, 68%). m.p. 98–100 °C,[1]H NMR (400 MHz, CDCl3) δ 6.78 (s, 1H), 5.89 (d, *J* = 9.8 Hz, 1H), 5.81 (dt, *J* = 9.8, 2.6 Hz, 1H), 4.87 (d, *J* = 1.3 Hz, 1H), 4.71–4.65 (m, 1H), 4.44 (d, *J* = 8.8 Hz, 1H), 4.33 (dd, *J* = 12.5, 6.8 Hz, 1H), 3.85–3.80 (m, 1H), 3.68–3.62 (m, 1H), 3.57–3.44 (m, 14H), 3.34–3.27 (m, 1H), 3.16–3.11 (m, 2H), 3.04–3.10 (m, 1H), 2.91–2.85 (t, *J* = 4.7 Hz, 3H), 2.91–2.85 (m, 1H), 2.65–1.72 (m, 22H), 1.62–1.14 (m, 20 H), 0.93–0.89 (m, 1H), 0.83 (t, *J* = 7.2 Hz, 3H).[13]C NMR (100 MHz, CDCl3) δ 202.92, 172.60, 147.69, 144.13, 129.35, 128.80, 103.41, 95.35, 82.25, 81.01, 80.70, 77.67, 73.26, 67.93, 66.13, 64.78, 64.46, 64.17, 62.90, 62.85, 61.04, 59.06, 57.75, 52.99, 49.43, 47.65, 47.59, 46.02, 44.96, 41.50, 41.17, 38.26, 37.37, 36.26, 35.43, 34.33, 34.19, 31.06, 30.98, 30.05, 28.44, 21.63, 20.59, 19.03, 17.83, 16.16, 9.40. HRMS-ESI: *m/z* calcd. for C47H78N3O11 [M + H]+, 860.5636, found 860.5627.

**13,14-dihydro spinosyn A (SPAH, 3).** To a solution of spinosyn A (100 mg, 0.137 mmol) in ethanol (5 mL), sodium borohydride (57 mg, 1.37 mmol) was added in portions over 30 min The mixture solution was stirred for 30 min, and then quenched by addition of saturated aqueous ammonium chloride (10 mL). The reaction mixture was diluted with water (10 mL) then extracted with ethyl acetate (3 × 20 mL). The combined organic layer was washed with brine, dried over anhydrous sodium sulfate, filtered and then the volatiles removed under vacuum. The residue was purified by silica gel column chromatography, eluting with 5% methanol in dichloromethane giving the 13,14-dihydro spinosyn A as colorless glass (57 mg, 57%). mp. 72–79 °C, [1]H NMR (400 MHz, CDCl3) δ: 5.87 (d, *J* = 9.6 Hz, 1H), 5.64 (dt, *J* = 9.6, 2.8 Hz, 1H), 4.87–4.86 (m, 2 H), 4.28–4.34 (m, 2H), 4.12–4.20 (m, 1H), 3.63–3.70 (m, 1H), 3.52–3.56 (m, 4 H), 3.48–3.49 (m, 9 H), 3.43–3.46 (m, 2H), 3.10 (t, *J* = 9.6 Hz, 1H), 2.63–2.69 (m, 1H), 2.46–2.56 (m, 3H), 2.20–2.30 (m, 9H), 1.50–2.10 (m, 9H), 1.33–1.49 (m, 5H), 1.24–1.32 (m, 8H), 1.13 (d, *J* = 6.8 Hz, 3H), 0.99–1.08 (m, 1H), 0.91–0.97 (m, 1H), 0.87 (t, *J* = 7.2 Hz, 3H); [13]C-NMR (100 MHz, CDCl3) δ 216.18, 172.05, 130.70, 128.55, 101.78, 95.43, 82.26, 81.07, 78.66, 77.68, 75.94, 75.79, 73.72, 67.87, 64.83, 60.95, 59.02, 57.68, 54.58, 54.53, 47.14, 47.02, 45.21, 44.47, 41.78, 41.42, 40.72, 37.68, 36.38, 35.68, 35.14, 32.95, 31.01, 29.33, 28.24, 21.60, 18.96, 18.34, 17.79, 11.98, 9.67; HRMS-ESI: *m/z* calcd for C41H68NO10 [M+H]+, 734.4843, found 734.4851.

**N-6-O-biotinylhexyl spinosyn B (Biotin-SPA, 4).** To a mixture of N-6-bromohexyl spinosyn B, **8** (76 mg, 0.086 mmol), potassium carbonate (12 mg, 0.087 mmol) in N, N-dimethylformamide (1 mL), D-Biotin (21 mg, 0.086 mmol) was added. The mixture was stirred for 24 h at room temperature. The mixture was then poured into water (15 mL) yielding the solid which was collected by filtration. The crude solid was further purified by flash column chromatography on silca gel with a mixed solvent of ethyl acetate and methanol (5:1, v/v) as an eluent giving N-6-O-biotinylhexyl spinosyn B (**4**) as colorless powder (52.7 mg, 58%). m.p 124–129 °C. [1]H NMR (500 MHz, CDCl3) δ: 6.77 (s, 1H), 5.87 (d, *J* = 10.0 Hz, 1H), 5.79 (dt, *J* = 10.0, 3.0 Hz, 1H), 5.73 (s, 1H), 5.47 (br, s, 1H), 4.84 (d, *J* = 1.5 Hz, 1H), 4.63–4.68 (m, 1H), 4.49–4.51 (dd, *J* = 7.5 Hz, 1H), 4.41 (d, *J* = 7.0 Hz, 1H), 4.28–4.32 (m, 2H), 4.05 (t, *J* = 7.0 Hz, 2H), 3.59–3.62 (m, 1H), 3.55 (s, 3H), 3.52–3.54 (m, 2H), 3.49 (s, 3H), 3.48 (s, 3H), 3.26–3.31 (m, 1H), 3.08–3.16 (m, 3H), 2.99–3.01 (m, 1H), 2.91 (dd, *J* = 12.5, 5 Hz, 1H), 2.84–2.88 (m, 1H), 2.74(d, *J* = 13 Hz, 1H), 2.36–2.41 (m, 2H), 2.32 (t, *J* = 7.5 Hz, 2H), 2.23–2.28 (m, 2H), 2.10–2.20 (m, 9H), 1.89–1.97 (m, 2H), 1.80–1.82 (m, 1H), 1.59–1.75(m, 7H), 1.52–1.59 (m, 3H), 1.38–1.49 (m, 8H), 1.29–1.36 (m, 6H), 1.27(d, *J* = 6.0 Hz, 3H), 1.24 (d, *J* = 6.0 Hz, 3H), 1.17(d, *J* = 7.5 Hz, 3H), 0.85–0.92 (m, 1H), 0.81 (t, *J* = 7.5 Hz, 3H); [13]C NMR (126 MHz, CDCl3) δ: 203.01, 173.88, 172.61, 163.68, 147.70, 144.08, 129.34, 128.77, 103.52, 95.42, 82.25, 80.99, 80.68, 77.68, 76.10, 73.65, 67.92, 64.56, 63.97, 61.95, 60.93, 60.15, 58.99, 57.68, 55.42, 53.92, 50.62, 49.41, 47.65, 47.58, 46.01, 41.52, 41.15, 40.51, 37.35, 37.21, 36.26, 34.34, 34.19, 33.29, 31.08, 30.09, 29.68, 28.64, 28.38, 28.22, 28.10, 26.83, 25.86, 24.76, 21.58, 19.33, 19.00, 17.77, 16.18, 9.34; HRMS-ESI: *m/z* calcd. for C56H90N3O13S, [M+H]+, 1044.6194, found 1044.6193.

**N-6-O-biotinylhexyl-13,14-dihydro spinosyn B (Biotin-SPAH, 5).** To a mixture of N-6-bromohexyl-13,14-dihydro spinosyn B, **9** (50 mg, 0.057 mmol) potassium carbonate (8 mg, 0.058 mmol) in N, N-dimethylformamide (1.5 mL), D-Biotin (14 mg, 0.057 mmol) was added. The mixture was stirred for 24 h at room temperature. The mixture was then poured into water (15 mL) giving white precipitate which was collected by filtration. The solid obtained was further purified by flash column chromatography on silca gel with a mixed eluent of ethyl acetate and methanol (5:1, v/v) giving oily N-6-O-biotinyl-hexyl-13,14-dihydro spinosyn B (**5**, 37.4 mg, 62%). [1]H NMR (500 MHz, CDCl3) δ: 5.88 (d, *J* = 9.5 Hz, 1H), 5.74 (br, s, 1H), 5.65 (dt, *J* = 9.5, 3.0 Hz, 1H), 5.41 (br, s, 1H), 4.87 (d, *J* =1.0 Hz, 1H), 4.84–4.86 (m, 1H), 4.53 (dd, *J* = 7.5, 5.0 Hz, 1H), 4.30–4.36 (m, 3H), 4.12–4.17 (m, 1H), 4.06 (t, *J* = 6.5 Hz, 2H), 3.63–3.66 (m, 1H), 3.56 (s, 3H), 3.52-3.55 (m, 2H), 3.51 (s, 3H), 3.50 (s, 3H), 3.46 (dd, *J* = 9.5, 3.5 Hz, 1H), 3.16–3.20 (m, 1H), 3.12 (t, *J* = 9.0 Hz, 1H), 2.93 (dd, *J* = 13.0, 5.0 Hz, 1H), 2.76 (d, *J* = 13 Hz, 1H), 2.67 (dd, *J* = 13.0, 3.0 Hz, 1H), 2.48–2.58 (m, 2H), 2.36–2.44 (m, 2H), 2.34 (t, *J* = 7.5 Hz, 2H), 2.22–2.30 (m, 3H), 2.18 (s, 3H), 2.01–2.13 (m, 4H), 1.91–2.02 (m, 2H), 1.81–1.85 (m, 2H), 1.56–1.76 (m,10H), 1.38–1.48 (m, 7H), 1.32–1.34 (m, 4H), 1.23–1.29 (m, 12H), 1.15 (d, *J* = 7.0 Hz, 3H), 1.01–1.08 (m,1H), 0.88 (t, *J* = 7.5 Hz,

3H). [13]C-NMR (126 MHz, CDCl3) δ: 216.19, 173.75, 172.12, 163.63, 130.68, 128.61, 101.93, 95.44, 82.27, 81.07, 78.73, 77.69, 75.97, 73.73, 67.89, 64.51, 63.85, 61.98, 60.98, 60.15, 59.04, 57.70, 55.37, 54.26, 47.13, 46.99, 45.28, 44.47, 41.80, 41.42, 40.54, 37.70, 36.99, 36.41, 35.76, 35.16, 33.91, 32.99, 31.21, 29.70, 29.31, 28.67, 28.37, 28.27, 28.24, 26.83, 25.88, 24.78, 21.66, 19.24, 18.99, 17.82, 12.23, 9.71. HRMS-ESI: *m/z* calcd. for C56H92N3O13S, [M+H]+, 1046.6351, found 1046.6354.

**N-demethyl spinosyn A (6).** To a solution of spinosyn A (5.00 g, 6.83 mmol) in methanol (200 mL), was added an aqueous sodium acetate solution (1.12 g NaOAc in 1 mL H2O) and Iodine (6.94 g, 27.32 mmol) successively. The mixture was refluxed for 12 h and 1 mol/L aqueous sodium hydroxide was added several times to keep the weak basic condition (pH = 8–9). After removal of methanol under vacuum, 1 M aqueous sodium thiosulfate (100 mL) was added and then extracted with ethyl acetate (3 × 100 mL). The combined organic layer was washed with saturated brine, dried over anhydrous sodium sulfate and concentrated under vacuum. The residue was purified by a silica column chromatography, eluting with a mixture of ethyl acetate, methanol and triethyl amine (100:10:1) to afford N-demethyl-spinosyn A (**6**) as a pale yellowish solid (3.96 g, yield 81%). mp. 81–89 °C, [1]H-NMR (400 MHz, CDCl3) δ: 6.78 (s, 1H), 5.89 (d, *J* = 9.8 Hz, 1H), 5.81 (d, *J* = 9.8 Hz, 1H), 4.89–4.84 (m, 1H), 4.73–4.64 (m, 1H), 4.53–4.47 (m, 1H), 4.36–4.29 (m, 1H), 3.69–3.62 (m, 1H), 3.59–3.55 (m, 4H), 3.53–3.45 (m, 10H), 3.35–3.26 (m, 1H), 3.17–3.09 (m, 2H), 3.06–2.99 (m, 1H), 2.92–2.84 (m, 1H), 2.51 (s, 3H), 2.45–2.32 (m, 3H), 2.31–2.23 (m, 1H), 2.23–2.12 (m, 2H), 2.02–1.90 (m, 2H), 1.83–1.71 (m, 1H), 1.62–1.40 (m, 9H), 1.35 (d, *J* = 6.1 Hz, 4H), 1.29 (d, *J* = 6.2 Hz, 3H), 1.19 (d, *J* = 6.7 Hz, 4H), 0.98–0.88 (m, 1H), 0.83 (t, *J* = 7.4 Hz, 3H); [13]C-NMR (100 MHz, CDCl3) δ: 202.76, 172.56, 147.57, 144.08, 129.33, 128.78, 103.12, 95.40, 82.24, 81.01, 80.85, 77.68, 76.05, 74.45, 67.91, 60.95, 60.59, 59.01, 57.70, 49.42, 47.58, 46.02, 41.50, 41.15, 37.36, 36.26, 34.29, 34.19, 33.11, 30.42, 30.11, 28.37, 27.35, 21.50, 19.04, 17.79, 16.24, 9.37; HRMS-ESI: *m/z* calcd. for C40H64NO10, [M+H]+, 718.4530, found 718.4524.

**N-3-chloro-2-hydroxypropyl spinosyn B (7).** To a solution of **6** (300 mg, 0.418 mmol) and methanol (20 mL), epichlorohydrin (131 μL, 1.67 mmol) was added. The mixture was stirred for 48 h at 50 °C. After evaporation of organic solvent under vacuum, the residue was purified by silica gel column chromatography, eluting with a mixed solvent of ethyl acetate and petroleum ether (3:1, v/v) giving N-3-chloro-2-hydroxypropyl spinosyn B (**7**) as a yellow glassy solid (206 mg, 62%). m.p. 89–91 °C,[1]H NMR (400 MHz, CDCl3) δ 6.78 (s, 1H), 5.89 (d, *J* = 10.0 Hz, 1H), 5.82 (dt, *J* = 10.0, 2.8 Hz, 1H), 4.85–4.90 (m, 1H), 4.66–4.72 (m, 1H), 4.45–4.51 (m, 1H), 4.33 (q, *J* = 7.5 Hz, 1H), 3.84–3.96 (m, 1H), 3.62–3.66 (m, 1H), 3.59–3.61 (m, 2H), 3.54–3.57 (m, 5H), 3.51–3.53 (m, 7H), 3.46–3.50 (m, 2H), 3.27–3.34 (m, 1H), 3.11–3.16 (m, 2H), 3.01–3.04 (m, 1H), 2.86–2.92 (m, 1H), 2.38–2.55 (m, 3H), 2.15–2.31 (m, 5H), 2.02–2.15 (m, 1H), 1.92–1.97 (m, 2 H), 1.70–1.85 (m, 2 H), 1.40–1.69 (m, 11H), 1.34–1.41 (m, 2H), 1.29–1.34 (m, 6H), 1.22–1.26 (m, 1H), 5.89 (d, *J* = 6.8 Hz, 3H) 0.83–0.95 (m, 1H), 0.84 (t, *J* = 7.5 Hz, 3H).[13]C NMR (100 MHz, CDCl3) δ 202.81, 172.53, 147.58, 144.12, 129.34, 128.80, 103.31, 95.45, 82.27, 81.05, 80.77, 77.71, 76.77, 67.94, 67.02, 66.72, 66.72, 66.35, 60.95, 59.02, 57.71, 56.82, 49.44, 47.61, 46.94, 46.03, 41.51, 41.17, 38.14, 37.38, 36.28, 34.32, 34.19, 30.98, 30.82, 28.39, 21.55, 20.85, 20.50, 19.02, 17.80, 16.18, 9.35.; HRMS-ESI: *m/z* calcd for C43H69ClNO11 [M + H]+, 810.4559, found 810.4564.

**N-6-bromohexyl spinosyn B (8).** The intermediate **6** (1000 mg, 1.39 mmol), potassium carbonate (385 mg, 1.39 mmol), and 1,6-dibromohexane (2.14 mL, 13.9 mmol) were mixed in acetonitrile (20 mL). The reaction mixture was stirred for 48 h at 45 °C. The reaction mixture was diluted with water (50 mL), and then extracted with ethyl acetate (3 × 20 mL). The combined organic layer was washed with brine, dried over anhydrous sodium sulfate and then filtered. After evaporation of organic solvent under vacuum, the residue obtained was purified by silica gel column chromatography using a mixed solvent of ethyl acetate and petroleum ether (1:1, v/v) as an eluent giving N-6-bromohexyl spinosyn B (**8**) as white solid (840 mg, 73%). m.p. 95–98 °C; [1]HNMR (500 MHz, CDCl3) δ: 6.74 (s, 1H), 5.84 (d, *J* = 9.5 Hz, 1H) 5.77 (dt, *J* = 9.5, 2.5 Hz, 1H), 4.82 (d, *J* = 1.5 Hz, 1H), 4.63 (m, 1H), 4.38 (d, *J* = 7.0 Hz, 1H), 4.28 (q, *J* = 7.0 Hz, 1H), 3.57–3.61 (m, 1H), 3.52 (s, 4H), 3.44–3.47 (m, 8H), 3.42–3.46 (m, 4H), 3.36–3.39 (m, 3H), 3.23–3.27 (m, 1H), 3.06–3.11 (m, 2H), 2.96–2.98 (m, 1H), 2.81–2.86 (m, 1H), 2.33–2.38 (m, 3H), 2.19–2.26 (m, 2H), 2.10–2.17 (m, 4H), 1.69–1.94 (m, 8H), 1.37–1.52 (m, 13H), 1.25 (d, *J* = 6.0 Hz, 3H), 1.21 (d, *J* = 6.5 Hz, 3H), 1.14 (d, *J* = 7.0 Hz, 3H), 0.83–0.90 (m, 1H), 0.78 (t, *J* = 7.5 Hz, 3H); [13]C–NMR (126 MHz, CDCl3) δ: 202.90, 172.53, 147.56, 144.09, 129.30, 128.78, 103.47, 95.41, 82.23, 81.01, 80.59, 77.66, 76.70, 76.05, 73.63, 67.90, 64.03, 60.91, 58.98, 57.67, 53.80, 50.63, 49.39, 47.64, 47.57, 45.60, 41.49, 41.14, 37.34, 37.23, 36.25, 34.31, 34.15, 33.91, 32.80, 31.09, 30.07, 28.38, 28.03, 26.27, 21.59, 19.29, 18.97, 17.77, 16.14, 9.33. HRMS-ESI: *m/z* calcd for C46H75BrNO10, [M+H]+, 880.4574, found 880.4572.

**N-6-bromohexyl-13,14-dihydro spinosyn B (9).** To a solution of N-6-bromohexyl spinosyn B (180 mg, 0.204 mmol) in ethanol (10 mL) under nitrogen atmosphere, sodium borohydride (78 mg, 2.04 mmol) was added in portions over 30 min The reaction progress was monitored by TLC until the starting material was disappeared. The reaction was stirred for 30 min at room temperature, and then quenched by addition of saturated aqueous ammonia chloride (10 mL). The

mixture was then diluted with water (10 mL) and extracted with ethyl acetate (20 × 20 mL). The organic layer was combined, dried over anhydrous sodium sulfate, filtered and evaporated under vacuum. The product was purified by chromatography on silca, eluting with a mixed solvent of petroleum and ethyl acetate (1:1, v/v), giving N-6-bromohexyl-13,14-dihydro-spinosyn B (9) as oil residue (101.6 mg, 56%). $^1$H NMR (500 MHz, CDCl$_3$) δ: 5.86 (d, $J = 10.0$ Hz, 1H), 5.62 (dt, $J = 10.0$, 3.0 Hz, 1H), 4.8–4.86 (m, 2H), 4.31 (q, $J = 6.5$ Hz, 1H), 4.27(dd, $J = 7.0$, 1.5 Hz, 1H), 4.10–4.15(m, 1H), 3.61–3.66 (m, 1H), 3.53 (s, 3H), 3.50–3.51 (m, 2H), 3.48–3.50 (m, 1H), 3.48 (s, 3H), 3.47 (s, 3H), 3.42–3.44 (m, 1H), 3.39 (t, $J = 7.0$ Hz, 2H), 3.09 (t, $J = 8.5$ Hz, 1H), 2.64 (dd, $J = 18$, 3.0 Hz, 1H), 2.44–2.55 (m, 3H), 2.33–2.39 (m, 2H), 2.18–2.28 (m, 3H), 2.15 (s, 3H), 2.0–2.14 (m, 3H), 1.78–1.91 (m, 8H), 1.69–1.78 (m, 1H), 1.51–1.63 (m, 4H), 1.35–1.45 (m, 8H), 1.26 (d, $J = 6.5$ Hz, 3H), 1.25 (d, $J = 6.0$ Hz, 3H), 1.18–1.23 (m, 3H), 1.12 (d, $J = 7.0$ Hz,3H), 0.98–1.05 (m, 1H), 0.86 (t, $J = 7.5$ Hz, 3H); $^{13}$C-NMR (126 MHz, CDCl$_3$) δ: 216.24, 172.10, 130.67, 128.56, 101.88, 95.42, 82.24, 81.03, 78.69, 77.66, 75.96, 73.77, 67.86, 63.98, 60.93, 59.00, 57.66, 54.68, 53.99, 50.66, 47.09, 46.95, 45.26, 44.45, 41.79, 41.40, 37.65, 37.08, 36.37, 35.72, 35.15, 33.96, 32.96, 32.79, 31.20, 29.66, 29.29, 28.23, 28.02, 27.99, 26.27, 21.61, 19.22, 18.96, 17.78, 9.66. HRMS-ESI: $m/z$ calcd for C$_{46}$H$_{77}$BrNO$_{10}$, [M+H]$^+$, 882.4731, found 882.4738.

**Statistical analysis**. All data are expressed as the mean ± s.d., and the number of samples is indicated in each figure legend. Results are representatives of at least three independent experiments. Inferential statistical analyses were performed with an unpaired two-tailed Student's $t$-test (*$P < 0.05$; **$P < 0.01$; ***$P < 0.001$). SPSS 18.0 or GraphPad Prism 8 was used for analysis.

**Reporting summary**. Further information on research design is available in the Nature Research Reporting Summary linked to this article.

## Data availability

All data supporting the findings of this study are available from the corresponding authors upon reasonable request. The atomic coordinates and structue factors of ASS1 have been deposited in the protein data bank, www.wwpdb.org (PDB code: 2NZ2). The uncropped gel or blot figures and original data underlying Figs. 1–6 and Supplementary Figs. S4–S8 are provided as a Source Data file. The synthesis routes and analytical spectra of the chemical derivatives presented in Figs. 1–2 are provided in the Supplementary Information document. The full mass spectrometry results of pull-down assay in excel form and raw metabolomic data are presented in Source date. Source data are provided with this paper.

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

## Acknowledgements

We thank Shaogang Liu, Jianping Liu, Fuliang Deng, and Shuyi Yu in Advanced Research Center of Central South University for the assistance in small molecule identification and metabolite quantification by HPLC-MS and Jun Zhou in Institute of Medical Science Research of Xiangya Hospital, Central South University for the help in confocal and immunohistochemical experiments. This study was supported by grants from China National Natural Science Foundation (81973710, 81673544, 81974469, 81672635, 81903107, 21472247), Hunan Provincial Key Program for Research and Development (2018SK21310, 2018SK21214), Outstanding Youth Foundation of Hunan Provincial Department of Education (20B589), Scientific Research Project of Hunan Provincial Health and Family Planning Commission (C2016125) and the postgraduate innovation project of Central South University of China (2016zzts167).

## Author contributions

Z.Z. and X.H. performed most cell, biochemical and animal experiments. N.X., L.C., and D.M. synthesized derivatives and probes under the guidance of S.L. X.C., W.L., T.L., J.J.L.,

M.W., F.K., K.P., Y.X., X.L., C.Y., O.L., and C.C. performed cell culture and animal experiments. Z.M. and W.Y. performed immunohistochemistry. L.X. contributed to MS analysis. D.C. performed the docking experiments. Z.Z. performed data analysis and wrote most of the manuscripts. S.L. wrote the chemistry part of the manuscript. J.L. guided part of this study and wrote part of the manuscripts. Z.L. initiated the project, led the project team, designed the experiments, and guided all aspects of this study.

## Competing interests

The authors declare no competing interests.
