## [Peer Review File · Nature Communications]

Reviewers' comments:

Reviewer #1 (Remarks to the Author):

Thank you for the opportunity to review the manuscript by Zou et al entitled "Naturally-occurring spinosyn A and its derivatives function as 2 argininosuccinate synthase activator and tumor inhibitor". In this manuscript they describe that ASS1 may be activated by a compound. This is in need of a major revision as it is a metabolism paper done using MTT assays which are not acceptable since there is a clear link of MTOR to arginine levels and Mitochondrial activity to arginine.

Major

1. Figure 1B and F: As arginine modulation has been shown to alter mitochondria oxygen consumption, MTT assays are NOT acceptable and are not acceptable in the metabolic cancer field. They authors should find an alternative method to demonstrate IC50s.
2. Figure 1G, the authors should demonstrate ASS1 expression of their input MDMBA231 cells, the xenografted cells and the treated cells. Also the authors don't take out their model long enough to see the shift of the curve. The authors should grow all tumors to at least 1000mm³. Finally, N=7 needs statistical justification as that is low. There statistics are driven by the 4 mice out of the 7 in the control arm. Also, since the animals were treated from day = 4 instead of at a consistent size. In addition, they should show tumor activity like Ki67 to show this is a proliferative effect. Also, since they demonstrate later in the paper that ASS1 isn't expressed in MDMBA231 cells, how do we know that this is a result of binding to the tumor in this mouse and not to an immune or other cell type?
3. Figure 2 – please find alternative for MTT assay.
4. Again for Figure 5, MTT assays are not acceptable.
5. The finding of a correlation between proliferation rates and ASS1 expression based on MTT is unexpected and should be correlated by multiple methods.
6. Figure 5E conclusion cannot be made by MTT assay.
7. Figure 5J is biased due to unequal amounts of tumors tested. This should be done with at least 100 cases in each arm that are age and ER/PR/Her2 matched.
8. Figure 6 is also based on MTT and needs to be redone and then interpreted.
9. The authors also haven't shown that they have an effect in the absence of arginine.
10. These experiments should also be done in vivo in the presence and absence of ASS1 in an immunocompetent mouse to see if there is an effect once macrophages are present.
11. By Seahorse, does their compound change oxygen consumption?

Reviewer #2 (Remarks to the Author):

In this manuscript, the authors found ASS1 served as a tumor suppressor in breast cancer, and its expression was reversely correlated to the ability of cell proliferation. Small molecules SPA and LM-2I could bind and activate ASS1, and therefore inhibit the production of pyrimidine and cell proliferation. The previous report had reported that decreased activity of ASS1 in cancers supports proliferation by activating CAD and facilitating pyrimidines synthesis, which was consistent with the present study. However, the mechanism can't fully explain the results showed in this MS. Detailed comments list below.

1. In figure 5e, overexpression of ASS1 in MDA-MB-231 decreased cell proliferation. In figure 1b, SPA and LM-2I both significantly inhibited breast cancer cell growth. However, in figure 6B, simultaneously application of ASS1 overexpression and SPA/ LM-2I unexpectedly promoted MDA-MB-231 cell growth. It's hard to understand by the explanation that overexpression of ASS1 decreased the sensitivity of MDA-MB-231 cells to the inhibition of SPA and LM-2I. Because increased activator will always enhance the enzyme activity. Similar problem with supplementary figure 6b should also be further studied.

2. Lots of studies had proved L-arginine is a conditionally essential basic amino acid primarily involved in DNA synthesis. Decreased ASS1 will affect the production of arginine and cell proliferation when deprivation of external arginine. To make sure the consequence of ASS1 deficiency in breast cancer cells, the authors should also detect the roles of SPA/ LM-21 on cell proliferation when depletion of arginine in the culture medium.

3. There are also some minor issues;

1) In page 6 line 113, “in a dose dependent manner (Fig.1b)”. However, fig 1b didn't show a dose dependent results.

2) In supplementary Fig.4f, xenograft's section from SPA/ LM-21 treatment group has much lesser tumor cells. The authors should explain this phenomenon.

Reviewer #3 (Remarks to the Author):

The manuscript by Zou et al describes the evaluation of antitumor activity of spinosyn A (SPA), the major component of the insecticide spinosad, and its semisynthetic derivative LM-21 in several breast cancer cell lines. To this end, the authors prepared the featured compounds and demonstrated that the compounds inhibit cancer cell proliferation by acting as upregulators of argininosuccinate synthase (ASS1) activity through a nucleophilic addition of a particular cysteine residue to the essential Michael acceptor moiety of SPA and LM-21. These results are nicely supported by data from enzymatic activity assays, validation of the binding kinetics of the compounds to ASS1 as well as site-directed mutagenesis and MS/MS analyses. Furthermore, the authors show that the compound-promoted activation of ASS1 inhibits breast cancer cell proliferation by blocking the biosynthesis of pyrimidines.

While the overall experiments seem to be well thought and carried out, we would like to note several critical points that need to be addressed by the authors. We fully support the publication of the manuscript if these points were to be addressed. Please find both major and minor points below.

Major points:

1. Figures 1b and 1e. The authors should include, at least in the Supplementary information, the dose response curves obtained for respective cell lines and the two tested compounds and not only show the IC50 values in the performed MTT viability assays. The authors should also correct the values on the y-axis in Fig. 1b and properly label the y-axis in Fig. 1e.

2. Figure 6b and Supplementary Figure 6b. Similar to Figure 1, the authors should include dose response curves in cell lines that overexpress ASS1 (Fig. 6b) as well as in ASS1-knockdown cells (Supp. Fig. 6b).

3. The authors should include the MS data that clearly and unambiguously identifies ASS1 as the 50 kDa protein from the biotin-SPA pulldown assay and not just enclose the identified peptides (as visible in Supp. Fig. 5c). The processed data should be presented as excel sheet in a manner that readers can interact with the data. Moreover, raw data should be uploaded to Pride.

Minor points:

1. Although the manuscript is comprehensibly written, the authors state in several occasions claims for which the data is not present. For example, the authors claim that they “then synthesized a series of derivative compounds of SPA and tested their inhibitory efficacy on cancer cells by MTT and colony formation assays” while only data for one compound (LM-21) is present in the manuscript. The authors similarly claim that “studies of relationship between structure of semi-synthesis SPA derivatives and activity indicated that the substitution at the nitrogen atom of 17-amino sugar group in SPA retained the biological activity” without supporting the claim with data or references to prior works. Could the authors elaborate why this data is missing from the manuscript?

2. A general question to the authors regarding the Cys mutagenesis experiments; Could you comment on why Cys97 was mutated into Asp while the other four Cys residues were all mutated into Ala?

3. The authors state in the Supplementary methods that commercially available spinosad was further purified by the method reported previously by their group but don't cite the aforementioned publication.

4. The authors should specify in the general information how the melting points of the prepared compounds have been determined.
5. We would recommend the authors to only reference the synthesis of N-demethyl spinosad (compound 6) as the procedure featured in this submission has already been published.

NCOMMS-20-08339-T

Responses to the Reviewer comments:

Reviewer #1:

This is in need of a major revision as it is a metabolism paper done using MTT assays which are not acceptable since there is a clear link of MTOR to arginine levels and Mitochondrial activity to arginine.

1. Figure 1B and F: As arginine modulation has been shown to alter mitochondria oxygen consumption, MTT assays are NOT acceptable and are not acceptable in the metabolic cancer field. They authors should find an alternative method to demonstrate IC50s.

3. Figure 2 – please find alternative for MTT assay.

4. Again for Figure 5, MTT assays are not acceptable.

5. The finding of a correlation between proliferation rates and ASS1 expression based on MTT is unexpected and should be correlated by multiple methods.

6. Figure 5E conclusion cannot be made by MTT assay.

8. Figure 6 is also based on MTT and needs to be redone and then interpreted.

Response: We thank the Reviewer very much for the constructive comments. As the Reviewer kindly suggested, we have repeated all experiments of Figure 1, 2, 5, and 6 by using crystal violet assay instead of MTT. The results from both MTT and crystal violet assay are parallely arranged and attached in this response. The results from both methods are very similar, and therefore the conclusion is unchanged. In the revised MS, The results from crystal violet assay are presented.

For crystal violet assay, cells were seeded in 96-well plates at a density of 2,000–5,000 cells per well. 12 h later cells were treated with chemicals (SPA, etc.) for 48 h, and then cells were washed with PBS and fixed in 4% PFA. Cells were then stained with 0.1% Crystal Violet (C0775, Sigma-Aldrich) for 20 min and washed with water. Cells were then incubated with 10% acetic acid for 20

min. The absorbance of extract was measured at 590 nm.

For MTT assay, cells were seeded in 96-well plates at a density of 2000 to 5000 cells per well. 12 h later cells were treated with chemicals (SPA, etc.) for 48 h. Cells were then incubated with MTT [3- (4, 5-dimethylthiazol-2-yl) -2, 5 diphenyltetrazolium bromide] (M2128, Sigma) for 4 h. Cells were solubilized in dimethyl sulfoxide (DMSO) and the absorbance was measured at 570 nm.

MTT assay

Crystal violet assay

Figure 1b

Figure 1e

Figure 1b

Figure 1e

The picture was not shown in previous version

The picture was not shown in previous version

Supplementary Figure 4a

Supplementary Figure 4b

Figure 2b

Supplementary Figure 5a

Figure 2b

Supplementary Figure 6a

Supplementary Figure 5b

Figure 2d

Supplementary Figure 6b

Figure 2d

MTT assay

Figure 5e

Crystal violet assay

Figure 5b, bottom

Figure 5c, bottom

Figure 6a

Supplementary Figure 6a

Supplementary Figure 7a

Figure 6a

Figure 6b

Supplementary Figure 7d

Figure 6d, bottom

Supplementary Figure 6b

Supplementary Figure 7c

Figure 6c, bottom

MTT assay

Figure 6c

Crystal violet assay

Supplementary Figure 7e

Figure 6e

Figure 6f

Figure 6g

Supplementary Figure 6e

Supplementary Figure 8c

Supplementary Figure 6f

Supplementary Figure 8d

2. Figure 1G, the authors should demonstrate ASS1 expression of their input MDMBA231 cells, the xenografted cells and the treated cells. Also the authors don't take out their model long enough to see the shift of the curve. The authors should grow all tumors to at least 1000mm³. Finally, N=7 needs statistical justification as that is low. There statistics are driven by the 4 mice our of the 7 in the control arm. Also, since the animals were treated from day = 4 instead of at a consistent size. In addition, they should show tumor activity like Ki67 to show this is a proliferative effect. Also, since they demonstrate later in the paper that ASS1 isn't expressed in MDMBA231 cells, how do we know that this is a result of binding to the tumor in this mouse and not to an immune or other cell

type?

Response: We thank the Reviewer very much for the constructive suggestions. We repeated the animal experiments treated with SPA and LM-2I. When the average tumor size reached about 100 mm³, the mice were randomly divided into 3 groups and treated with vehicle, SPA (10 mg/kg/day), or LM-2I (5 mg/kg/day) , *i.p.* every other day. Tumor size and mouse body weight were measured every two days. When the tumor size in the vehicle-treated group exceeded 1000 mm³ (28 days after treatment), mice were sacrificed. the tumors were dissected and weighed (Fig. 1g-i). Ki-67 expression in tumors was detected by IHC (Supplementary Fig. 5c). MDA-MB-231 cells and xenografted tumors express ASS1 detected by western blot (Fig. 5a and Fig. 6b) and IHC (Supplementary Fig. 5b) , although at lower level as compared with other breast cancer cell lines, such as MCF-7 (Fig. 5a) .

7. Figure 5J is biased due to unequal amounts of tumors tested. This should be done with at least 100 cases in each arm that are age and ER/PR/Her2 matched.

Response: We thank the Reviewer very much for the constructive suggestions. We analyzed more breast cancer tissue samples from patients with follow-up data that are age and ER/PR/Her2 matched (Fig. 5h and Supplementary Table 1).

9. The authors also haven't shown that they have an effect in the absence of arginine.

Response: We thank the Reviewer very much for the comments. We analyzed the proliferation rates of cells treated with SPA and LM-2I in medium with or without arginine. We found that although cells cultured in arginine-depleted medium grew slowly they were still sensitive to SPA and LM-2I treatment (Attached Figure 1).

Attached Figure 1 | The comparison of the effect of arginine deprivation on the sensitivity of breast cancer cells to SPA/LM-2I treatment. Crystal violet assay for the viability of MDA-MB-231 (a, b) and MCF-7 (c, d) cells treated with either SPA (a, c) or LM-2I (b, d) in normal or arginine-depleted medium for 48 h. Data represent mean \pm s.d., * $P < 0.05$, ** $P < 0.01$, *** $P < 0.001$.

10. These experiments should also be done in vivo in the presence and absence of ASS1 in an immunocompetent mouse to see if there is an effect once macrophages are present.

Response: We thank the Reviewer very much for the comments. We apologize for the confusing. To confirm that SPA and LM-2I -induced tumor inhibition is indeed via targeting ASS1, we established ASS1 stable knockout (ASS1-KO) MDA-MB-231 cells by using CRISPR/cas9 system. ASS1-KO and control MDA-MB-231 cells were treated with different concentrations of SPA or LM-2I, and the cell viability was measured by crystal violet assay. We found that ASS1-KO MDA-MB-231 cells did not respond or were very insensitive to the treatment of SPA (Supplementary Fig. 7b) and LM-2I (Fig. 6b), suggesting that the

anticancer activity of SPA and LM-2I is indeed through targeting ASS1.

Nu/nu mice were used in our animal studies (Fig. 1g-i and Fig. 5e-f). Macrophages are presented in Nu/nu mice [Vetvicka, V. et al. Macrophages of athymic nude mice: Fc receptors, C receptors, phagocytic and pinocytic activities. *Eur J Cell Biol* **35**, 35-40 (1984)].

11. By Seahorse, does their compound change oxygen consumption?

Response: We thank the Reviewer very much for the comments. Seahorse analysis was performed to detect the oxygen consumption rate (OCR) in MDA-MB-231 cells treated with LM-2I, and we found that treatment with LM-2I did not significantly change the OCR.

Attached Figure 2 | The effect of LM-2I on cell oxygen consumption rate.

Oxygen consumption rate (OCR) was measured by Seahorse in MDA-MB-231 cells treated with LM-2I for 24 h. Data represent mean \pm s.d..

Reviewer #2 (Remarks to the Author):

In this manuscript, the authors found ASS1 served as a tumor suppressor in breast cancer, and its expression was reversely correlated to the ability of cell proliferation. Small molecules SPA and LM-2I could bind and activate ASS1, and therefore inhibit the production of pyrimidine and cell proliferation. The previous report had reported that decreased activity of ASS1 in cancers supports proliferation by activating CAD and facilitating pyrimidines synthesis,

which was consistent with the present study. However, the mechanism can't fully explain the results showed in this MS. Detailed comments list below.

1. In figure 5e, overexpression of ASS1 in MDA-MB-231 decreased cell proliferation. In figure 1b, SPA and LM-2I both significantly inhibited breast cancer cell growth. However, in figure 6B, simultaneously application of ASS1 overexpression and SPA/LM-2I unexpectedly promoted MDA-MB-231 cell growth. It's hard to understand by the explanation that overexpression of ASS1 decreased the sensitivity of MDA-MB-231 cells to the inhibition of SPA and LM-2I. Because increased activator will always enhance the enzyme activity. Similar problem with supplementary figure 6b should also be further studied.

Response: We thank the Reviewer very much for the comments. We apologize for the confusing. To confirm that SPA and LM-2I -induced tumor inhibition is indeed via targeting ASS1, we established ASS1 stable knockout (ASS1-KO) MDA-MB-231 cells by using CRISPR/cas9 system. ASS1-KO and control MDA-MB-231 cells were treated with different concentrations of SPA or LM-2I, and the cell viability was measured by crystal violet assay. We found that ASS1-KO MDA-MB-231 cells did not respond or were very insensitive to the treatment of SPA (Supplementary Fig. 7b) and LM-2I (Fig. 6b), suggesting that the anticancer activity of SPA and LM-2I is indeed through targeting ASS1.

Overexpression of ASS1 (ASS1-OE) in MDA-MB-231 decreased cell proliferation rate (Fig 5b) while knockdown of ASS1 (ASS1-KD) in MCF-7 cells increased cell proliferation rate (Fig. 5c). Interestingly, as compared with their controls, ASS1-OE cells are less sensitive to while ASS1-KD cells are more sensitive to SPA/LM-2I treatment (Fig. 6c-d and Supplementary Fig. 7c-d), suggesting that cancer cells with lower level of ASS1 expression are more sensitive to SPA and LM-2I.

It is worth noting that the value presented in Fig. 6b-d and Supplementary Fig. 7b-d is the relative value of treated to untreated cells, and the value of all untreated (control, KD, OE, KO) cells is defined as 1 (100%).

2. Lots of studies had proved L-arginine is a conditionally essential basic amino acid primarily involved in DNA synthesis. Decreased ASS1 will affect the production of arginine and cell proliferation when deprivation of external arginine. To make sure the consequence of ASS1 deficiency in breast cancer cells, the authors should also detect the roles of SPA/ LM-2I on cell proliferation when depletion of arginine in the culture medium.

Response: We thank the Reviewer very much for the suggestions. We analyzed the proliferation rates of cells treated with SPA and LM-2I in medium with or without arginine. We found that although cells cultured in arginine-depleted medium grew slowly they were still sensitive to SPA and LM-2I treatment (Attached Figure 1).

Attached Figure 1 | The comparison of the effect of arginine deprivation on the sensitivity of breast cancer cells to SPA/LM-2I treatment. Crystal violet assay for the viability of MDA-MB-231 (a, b) and MCF-7 (c, d) cells treated with either SPA (a, c) or LM-2I (b, d) in normal or arginine-depleted medium for 48 h. Data represent mean \pm s.d., * $P < 0.05$, *** $P < 0.01$, **** $P < 0.001$.

3. There are also some minor issues;

1) In page 6 line 113, "in a dose dependent manner (Fig.1b)". However, fig 1b didn't show a dose dependent results.

Response: We thank the Reviewer very much for the suggestions. The dose-dependent response curves of breast cancer cells to SPA and LM-21 are presented in Supplementary Fig.4a and 4b.

2) In supplementary Fig.4f, xenograft's section from SPA/LM-21 treatment group has much lesser tumor cells. The authors should explain this phenomenon.

Response: We thank the Reviewer very much for the comments. The xenograft tumors treated by SPA/LM-21 had less density of tumor cells (Supplementary Fig.5a). This may be due to less cell proliferation (Supplementary Fig. 5c) in tumors treated with SPA/LM-21 as compared with those treated with vehicles.

Reviewer #3 (Remarks to the Author):

The manuscript by Zou et al describes the evaluation of antitumor activity of spinosyn A (SPA), the major component of the insecticide spinosad, and its semisynthetic derivative LM-21 in several breast cancer cell lines. To this end, the authors prepared the featured compounds and demonstrated that the compounds inhibit cancer cell proliferation by acting as upregulators of argininosuccinate synthase (ASS1) activity through a nucleophilic addition of a particular cysteine residue to the essential Michael acceptor moiety of SPA and LM-21. These results are nicely supported by data from enzymatic activity assays, validation of the binding kinetics of the compounds to ASS1 as well as site-directed mutagenesis and MS/MS analyses. Furthermore, the authors show that the compound-promoted activation of ASS1 inhibits breast cancer cell proliferation by blocking the biosynthesis of pyrimidines.

While the overall experiments seem to be well thought and carried out, we would like to note several critical points that need to be addressed by the authors. We fully support the publication of the manuscript if these points were to be addressed. Please find both major and minor points below.

Major points:

1. Figures 1b and 1e. The authors should include, at least in the Supplementary information, the dose response curves obtained for respective cell lines and the two tested compounds and not only show the IC50 values in the performed MTT viability assays. The authors should also correct the values on the y-axis in Fig. 1b and properly label the y-axis in Fig. 1e.

Response: We thank the Reviewer very much for the constructive suggestions. We correct the values on the y-axis in Fig. 1b and the label of y-axis in Fig. 1e. The dose-response curves of different cell lines to SPA and LM-2I are presented in Supplementary Fig.4a-b.

2. Figure 6b and Supplementary Figure 6b. Similar to Figure 1, the authors should include dose response curves in cell lines that overexpress ASS1 (Fig. 6b) as well as in ASS1-knockdown cells (Supp. Fig. 6b).

Response: We thank the Reviewer very much for the constructive suggestions. The dose-response curves of different cell lines (control and ASS1-OE MDA-MB-231 cells; control and ASS1-KD MCF-7 cells; control and ASS1-KO MDA-MB-231 cells) to SPA and LM-2I are presented in Fig. 6b-d and Supplementary Fig.7b-d.

3. The authors should include the MS data that clearly and unambiguously identifies ASS1 as the 50 kDa protein from the biotin-SPA pulldown assay and not just enclose the identified peptides (as visible in Supp. Fig. 5c). The processed data should be presented as excel sheet in a manner that readers

can interact with the data. Moreover, raw data should be uploaded to Pride.

Response: We thank the Reviewer very much for the constructive suggestions.

The full mass spectrometry results in excel form are presented in Source data Figure S6c and raw data was named Biotin-SPA pull-down.raw.

Minor points:

1. Although the manuscript is comprehensibly written, the authors state in several occasions claims for which the data is not present. For example, the authors claim that they “then synthesized a series of derivative compounds of SPA and tested their inhibitory efficacy on cancer cells by MTT and colony formation assays” while only data for one compound (LM-21) is present in the manuscript. The authors similarly claim that “studies of relationship between structure of semi-synthesis SPA derivatives and activity indicated that the substitution at the nitrogen atom of 17-amino sugar group in SPA retained the biological activity” without supporting the claim with data or references to prior works. Could the authors elaborate why this data is missing from the manuscript?

Response: We thank the Reviewer very much for the comments. In addition to LM-21, the data related to synthesis of other derivative compounds of SPA and their inhibitory efficacy on cancer cells are included in an ongoing patent application. The reference from our previous work [Liu, S.Y. et al. Application of pleocidin derivatives in preparing antitumor agent and anti-KSHV. CN 201610355188.0. (China, 2016)] is cited after “We then synthesized a series of derivative compounds of SPA and tested their inhibitory efficacy on cancer cells by crystal violet staining and colony formation assays” and “Studies of the relationship between structure of semi-synthesis SPA derivatives and activity indicated that the substitution at the nitrogen atom of 17-amino sugar group in SPA retained the biological activity”.

2. A general question to the authors regarding the Cys mutagenesis

experiments; Could you comment on why Cys97 was mutated into Asp while the other four Cys residues were all mutated into Ala?

Response: We thank the Reviewer very much for the comments. At the beginning of this study, all Cys were mutated to Ala, however, the protein with Cys97-to-Ala mutation had very low solubility. According to the report by Liu et al., we changed the Cys97 to Asp, which improved the solubility of the protein. [Liu, C.X. et al. Adenanthin targets peroxiredoxin I and II to induce differentiation of leukemic cells. *Nat Chem Biol* **8**, 486-93 (2012)]

3. The authors state in the Supplementary methods that commercially available spinosad was further purified by the method reported previously by their group but don't cite the aforementioned publication.

Response: We thank the Reviewer very much for the comments. The reference is cited. [Mei, G. et al. Study on Purification of Spinosad A from Spinosad. *Fine Chemical Intermediates* **43**, 14-16 (2013)]

4. The authors should specify in the general information how the melting points of the prepared compounds have been determined.

Response: We thank the Reviewer very much for the suggestions. The melting points were determined on an X-4 Digital micro-melting point apparatus without correction.

5. We would recommend the authors to only reference the synthesis of N-demethyl spinosad (compound 6) as the procedure featured in this submission has already been published.

Response: We thank the Reviewer very much for the suggestions. We cite the references for the synthesis schemes of N-demethyl spinosyn A. [Ma, D.Y. et al. Synthesis and antiproliferative activities of novel quaternary ammonium spinosyn derivatives. *Bioorg Med Chem Lett* **28**, 3346-3349 (2018)].

Reviewers' comments:

Reviewer #1 (Remarks to the Author):

I appreciate your efforts and have no further comments.

Reviewer #2 (Remarks to the Author):

In this revised manuscript, the authors had well addressed all the critiques raised by the reviewers by providing several new data panels and literature review. The manuscript is now suitable for publication in NC.

Reviewer #3 (Remarks to the Author):

The authors clarified all open points in a convincing manner, and added data where needed. Thus, my initial (minor) concerns are sufficiently addressed and I recommend publication.